# Phase transition and remodeling complex assembly are important for SS18-SSX oncogenic activity in synovial sarcomas

Yanli Cheng [1], Zhongtian Shen [1], Yaqi Gao[1], Feilong Chen[1], Huisha Xu[1], Qinling Mo[1], Xinlei Chu[2], Chang-liang Peng[3], Takese T. McKenzie [4], Bridgitte E. Palacios [4], Jian Hu [4], Hao Zhou [1✉] & Jiafu Long [1,5✉]

Oncoprotein SS18-SSX is a hallmark of synovial sarcomas. However, as a part of the SS18-SSX fusion protein, SS18's function remains unclear. Here, we depict the structures of both human SS18/BRG1 and yeast SNF11/SNF2 subcomplexes. Both subcomplexes assemble into heterodimers that share a similar conformation, suggesting that SNF11 might be a homologue of SS18 in chromatin remodeling complexes. Importantly, our study shows that the self-association of the intrinsically disordered region, QPGY domain, leads to liquid-liquid phase separation (LLPS) of SS18 or SS18-SSX and the subsequent recruitment of BRG1 into phase-separated condensates. Moreover, our results show that the tyrosine residues in the QPGY domain play a decisive role in the LLPS of SS18 or SS18-SSX. Perturbations of either SS18-SSX LLPS or SS18-SSX's binding to BRG1 impair NIH3T3 cell transformation by SS18-SSX. Our data demonstrate that both LLPS and assembling into chromatin remodelers contribute to the oncogenic activity of SS18-SSX in synovial sarcomas.

[1] State Key Laboratory of Medicinal Chemical Biology, Tianjin Key Laboratory of Protein Science, and College of Life Sciences, Nankai University, 94 Weijin Road, 300071 Tianjin, China. [2] Department of Epidemiology and Biostatistics, Tianjin Medical University Cancer Institute and Hospital, 300060 Tianjin, China. [3] Department of Orthopaedics, The Second Hospital, College of Medicine, Shandong University, 250033 Jinan, China. [4] Department of Cancer Biology, The University of Texas MD Anderson Cancer Center, The University of Texas MD Anderson Cancer Center UT Health Graduate School of Biomedical Sciences, Houston, TX 77030, USA. [5] Nankai International Advanced Research Institute (Shenzhen Futian), 518045 Shenzhen, Guangdong, China. ✉email: haozhou@nankai.edu.cn; jflong@nankai.edu.cn

Synovial sarcoma (SyS) is a malignant neoplasm that accounts for 10–20% of all soft-tissue tumors with a poor prognosis in young adults[1]. The hallmark genetic feature of SyS is the recurrent and specific chromosomal translocation, t(X;18)(p11.2;q11.2), in which the *SS18* gene on chromosome 18 is fused to one of the three closely related genes on the X chromosome, *SSX1*, *SSX2*, and rarely *SSX4*[2–4], resulting in an in-frame fusion gene *SS18-SSX*. This remarkable translocation is present in virtually 100% of synovial sarcomas and is often the only cytogenetic aberration[1]. Contrasting with conventional translocations in other soft-tissue sarcomas, the oncofusion protein SS18-SSX lacks a DNA binding domain and is thought to exert its activity by combining with other chromatin modifiers[5].

The ATP-dependent chromatin remodeling complex BAF, also known as mammalian SWI/SNF complex, is a multi-subunit remodeler and is crucial for regulating gene expression and developmental programming. The misregulation of the BAF complex leads to neurological disorders and human malignancies[6]. Moreover, SS18, as a part of the SS18-SSX chimera, is an integral subunit of the canonical BAF complex (CBAF) by binding to the CBAF complex through interaction with the catalytic subunit BRG1 or BRM[7–10]. The recently obtained architecture of the human CBAF complex has provided some structural information about the complex assembly and chromatin remodeling, however, structural information on SS18 in the assembled CBAF complex is limited[11,12]. Notably, it is specific to SyS that the competition between SS18 and SS18-SSX for assembly within the CBAF complex leads to a biochemically aberrant complex and thereby disrupts gene expression[13,14].

It has been reported that deletion of N-terminal 181 amino acids of the oncoprotein SS18-SSX1 causes the loss of its transforming activity[15]. This observation, together with the results showing that SS18 or SSX overexpression alone does not generate tumors, implicated that both partners of SS18-SSX play important roles in synovial sarcomagenesis[13,15]. Moreover, SS18 or SS18-SSX features a low-complexity sequence domain (LCD), which is rich in glutamine, proline, glycine, and tyrosine (the QPGY domain) and is important for transcriptional activity[15,16]. Notably, the intrinsically disordered LCDs of oncofusion FUS/EWS/ TAF15 (FET) protein family were reported to involve the formation of dynamic protein condensates through a physical process known as liquid–liquid phase separation (LLPS) that controls gene transcription[17–19]. Collectively, it is of great importance in testing the role of SS18 in the oncofusion SS18-SSX1 through its binding to the CBAF complex or the QPGY domain in the occurrence and development of SyS. In this study, we report the crystal structures of the human SS18/BRG1 heterodimer derived from mammalian CBAF complex and the yeast SNF11/SNF2 heterodimer derived from *S. cerevisiae* SWI/SNF complex, with a resolution of 2.39 and 2.15 Å, respectively. In addition, our results reveal that the LCD of SS18 or SS18-SSX (QPGY domain) can lead to LLPS through tyrosine residues-mediated self-association. Our study suggests that phase separation of SS18-SSX and the binding of SS18-SSX to chromatin remodeling complex are important for the transformation activity of the oncoprotein SS18-SSX.

## Results

### Crystal structure of the complex of BRG1 and SS18 or SS18-SSX1.
SS18-SSX1 oncofusion protein is generated by substituting the eight-extreme carboxyl-terminal residues of SS18 (aa 379–387) with a carboxyl-terminal 78-residue fragment of SSX1 (aa 111–188) (Fig. 1a). It was reported that either SS18 or SS18-SSX1 can bind to the N-terminal region of the ATPase subunit BRG1 or BRM of chromatin remodeling complex[9,10]. However, the structural

information of SS18 binding to the remodeling complex remains limited in reported structures[11,12]. Accordingly, we initially confirmed the interaction between an N-terminal 282-residue fragment of BRG1 (aa 1–282, Brg1$^{(1-282)}$) and SS18, herein named BRG1$^{(1-282)}$/SS18, ("/" denotes protein complexes with separate chains and similar structures hereafter) by showing that the two proteins coeluted from a size-exclusion column (Supplementary Fig. 1a). Next, we mapped each binding region of BRG1 and SS18 using a truncation-based method combined with size-exclusion chromatography (Supplementary Fig. 1b–d). Notably, BRG1$^{(172-213)}$ and SS18$^{(14-101)}$ assembled into a complex in an analytical size-exclusion column (black line in Supplementary Fig. 1c and SDS-PAGE in Supplementary Fig. 1d). Analytical ultracentrifugation further confirmed that the BRG1$^{(172-213)}$/ SS18$^{(14-101)}$ complex forms a heterodimer with a stoichiometry of 1:1 and molecular mass of 14.7 kDa (black line in Supplementary Fig. 1f). Given these results, we concluded that the BRG1/SS18 or BRG1/SS18-SSX1 subcomplex forms a heterodimer through the interaction between fragments BRG1$^{(172-213)}$ and SS18$^{(14-101)}$.

To understand how BRG1 and SS18, or SS18-SSX1, bind to each other, we attempted to determine the crystal structure of the heterodimer BRG1$^{(172-213)}$/SS18$^{(14-101)}$. We succeeded in obtaining crystals of the single polypeptide created by the fusion of BRG1$^{(172-213)}$ to the N-terminus of SS18$^{(14-101)}$ with a tobacco etch virus (TEV)-cleavable segment. The purified single-chain fusion protein of BRG1$^{(172-213)}$ and SS18$^{(14-101)}$, herein named BRG1$^{(172-213)}$-SS18$^{(14-101)}$, ("-" denotes proteins in a single-chain fusion, similar structures hereafter) was eluted as a single peak from an analytical size-exclusion column (red line in Supplementary Fig. 1c and SDS-PAGE in Supplementary Fig. 1e) and assembled into a heterodimer with a molecular mass of 15.8 kDa from the sedimentation velocity (SV) experiment (red line in Supplementary Fig. 1f). The crystal structure of BRG1$^{(172-213)}$-SS18$^{(14-101)}$ was determined at a resolution of 2.39 Å with four copies of the complex molecule in one asymmetric unit (Supplementary Table 1). In the final model, BRG1$^{(172-213)}$ was resolved from aa 172 to 207, including the QLQ domain (Fig. 1b and Supplementary Fig. 2). SS18$^{(14-101)}$ was well-resolved from aa 14 to 79, including the SNH domain (Fig. 1b and Supplementary Fig. 3). The overall structure of BRG1$^{(172-213)}$-SS18$^{(14-101)}$ resembles a four-helix bundle, which span residues 174–190 (αA), 198–205 (αB) in BRG1$^{(172-213)}$ and residues 19–39 (α1), 44–74 (α2) in SS18$^{(14-101)}$ (Fig. 1b and Supplementary Figs. 2 and 3). The interaction between BRG1$^{(172-213)}$ and SS18$^{(14-101)}$ was maintained through hydrogen-bonding and hydrophobic interactions. The hydrophobic amino acids (I32, L54, and A65) on SS18 and the amino acids at the corresponding positions of BRG1 form three groups of hydrophobic cores (Fig. 1c–e).

To validate the interactions observed in the structure of the BRG1$^{(172-213)}$–SS18$^{(14-101)}$ complex, we performed a series of mutagenesis studies. Residues A65, L54, and I32 were mutated to glutamic acid in SS18 or SS18-SSX1, herein referred to as SS18(3M) and SS18(3M)-SSX1, respectively (Fig. 1c–e). As anticipated, co-immunoprecipitation (co-IP) analyses showed that the 3M mutations in SS18 or SS18-SSX1 can largely disrupt the binding of SS18 (lane 3 compared to lane 2 in Fig. 1f) or SS18-SSX1 (lane 4 compared to lane 2 in Supplementary Fig. 4a) to BRG1. We used circular dichroism to confirm similar behavior between wild-type (WT) and mutant SS18$^{(14-101)}$, which ensured that any loss in BRG1 binding activity was not due to decreased SS18$^{(14-101)}$ protein stability (Supplementary Fig. 1g). In addition, according to the hydrogen bond interactions between BRG1$^{(172-213)}$ and SS18$^{(14-101)}$, residues Q176 and Q183 of BRG1 were substituted with alanine (Supplementary Fig. 1h). Remarkably, either the double (Q176A, Q183A) or the single mutations (Q183A) almost completely abolished

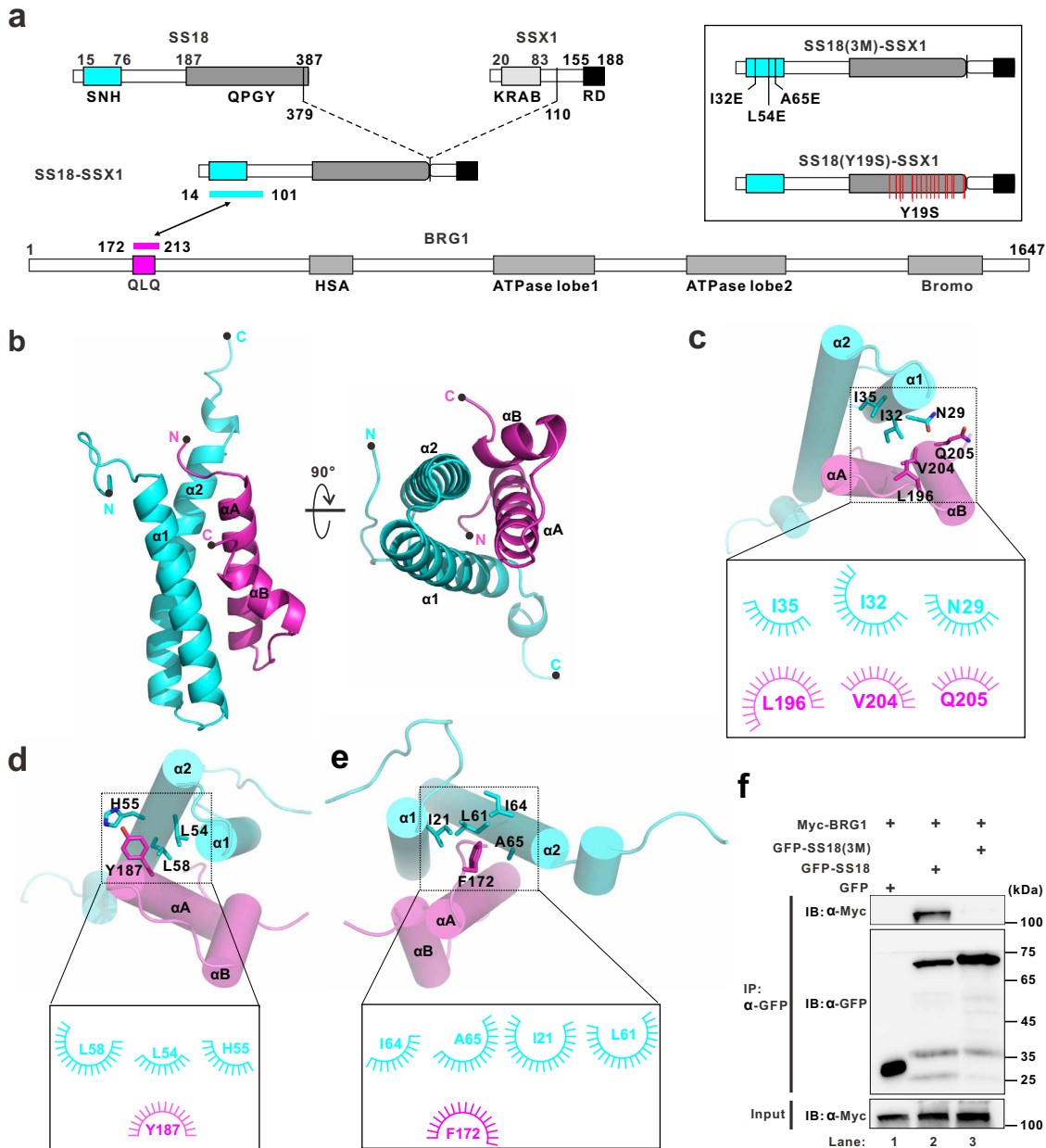

**Fig. 1 Crystal structure of the complex BRG1 and SS18 or SS18-SSX1. a** Schematic representation of full-length SS18, SSX1, SS18-SSX1, and BRG1. The oncoprotein SS18-SSX1 encompasses 457 amino acids and preserves the amino-terminal 379 amino acids of SS18 as well as the carboxy-terminal 78 amino acids of SSX1. The protein fragments of the SS18$^{(14-101)}$/BRG1$^{(172-213)}$ complex used for structural determination are indicated by a two-way arrow, and colored cyan and magenta, respectively. The schematic diagram in the insert indicates the mutant SS18(3M)-SSX1, which contained three amino acids I32, L54, and A65 mutated to glutamic acid in the SNH domain, and the mutant SS18(Y19S)-SSX1, which contained 19 tyrosine residues mutated to serine in the QPGY domain. **b** Cartoon representation of the SS18$^{(14-101)}$ (cyan)/BRG1$^{(172-213)}$ (magenta) complex viewed from the side and bottom. The N- and C-termini of the two proteins are labeled. PDB entry code: 7VRB. **c**–**e** Ligplot diagrams in the black frame indicate hydrophobic interactions between SS18 and BRG1. Three groups of hydrophobic cores are shown as spoked arcs, respectively in (**c**–**e**). **f** Co-IP experiments testing the interaction between SS18 wild-type (WT) or the mutant SS18(3M) and BRG1. The mutant SS18(3M) contains three amino acid mutations I32E, L54E, and A65E. Extracts were prepared from HEK293T cells transfected with combinations of plasmids, as indicated. The bottom panel shows 3% of the Myc-Brg1 as input for each IP.

the binding of BRG1 to SS18 (Supplementary Fig. 1i). Collectively, these results confirmed the interaction mode revealed by the structure of BRG1$^{(172-213)}$-SS18$^{(14-101)}$.

**SNF11 and SNF2 have a similar binding model to that of SS18 and BRG1.** Prior studies have reported structures of the yeast SWI/SNF complex[20,21]. However, there is limited structure information on SNF11, as it is an indispensable part of the complex[22].

According to the sequence alignment, we found that the yeast SNF11 was highly conserved to the SNH domain of human SS18, which is involved in binding to the QLQ domain of BRG1 (Fig. 1 and Supplementary Fig. 3). This observation, together with the sequence alignment showing that QLQ domains of BRG1 and SNF2, the yeast homolog of BRG1, are highly conserved, suggests that SNF11 might assemble into the SWI/SNF complex through binding to the QLQ domain of SNF2 (Supplementary Fig. 2).

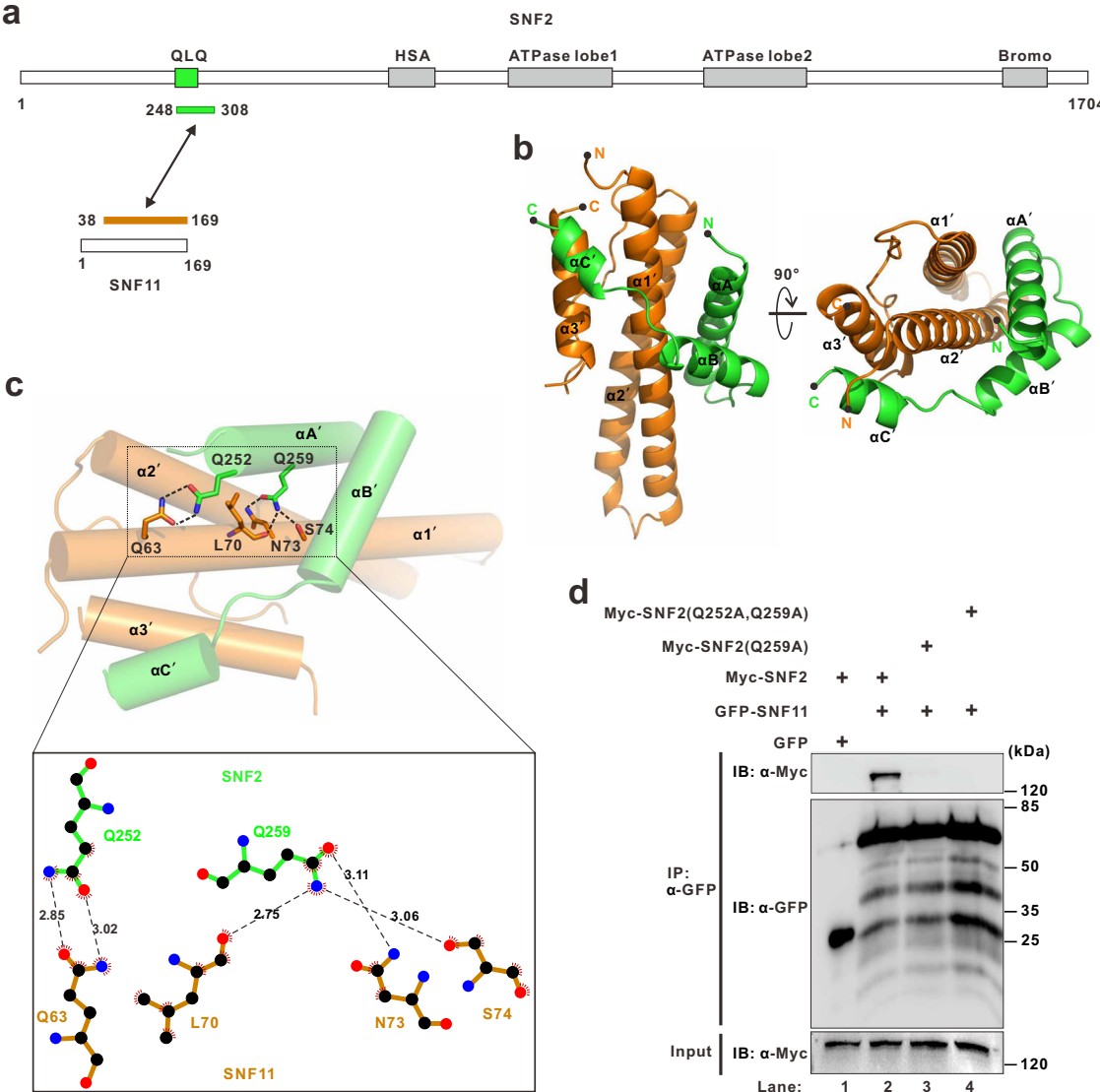

**Fig. 2 Crystal structure of complex SNF11 and SNF2. a** Schematic representation of full-length SNF11 and SNF2. The protein fragments of the SNF11$^{(38-169)}$/SNF2$^{(248-308)}$ complex used for structural determination are indicated by a two-way arrow, and colored orange and green, respectively. **b** Cartoon representation of the SNF11$^{(38-169)}$ (orange)/SNF2$^{(248-308)}$ (green) complex viewed from the side and bottom. The N- and C-termini of the two proteins are labeled. PDB entry code: 7VRC. **c** Ligplot diagram in the black frame indicates hydrogen-bonding interactions between SNF11 and SNF2. Hydrogen bonds are shown as black dotted lines. The numbers above the lines represent distance and the unit is Å. Black solid dots represent carbon atoms, blue solid dots represent nitrogen atoms, and red solid dots represent oxygen atoms. **d** Co-IP experiments testing the interaction between SNF2 wild-type (WT) or the mutant SNF2 and SNF11. Extracts were prepared from HEK293T cells transfected with combinations of plasmids, as indicated. The bottom panel shows 3% of the Myc-SNF2 as input for each IP.

Therefore, we coexpressed SNF11 and a 60-residue fragment, including the QLQ domain of SNF2 (aa 248–308, SNF2$^{(248-308)}$), and confirmed that SNF11 and SNF2$^{(248-308)}$ assembled into a heteromeric complex as indicated by their coelution from an analytical size-exclusion column (Supplementary Fig. 5a). Next, we mapped the SNF2-binding region on SNF11 to a 132-residue fragment (aa 38-169, SNF11$^{(38-169)}$; Fig. 2a and Supplementary Fig. 5b). Analytical ultracentrifugation further confirmed that the SNF11$^{(38-169)}$/SNF2$^{(248-308)}$ complex formed a heterodimer with a stoichiometry of 1:1 and molecular mass of 20.0 kDa (Supplementary Fig. 5c). Given these results, we concluded that the SNF11 subunit assembles into the SWI/SNF complex by forming a heterodimer through the interaction between fragments SNF11$^{(38-169)}$ and SNF2$^{(248-308)}$.

To reveal the binding model of SNF11 and SNF2, we determined the crystal structure of the SNF11$^{(38-169)}$/SNF2$^{(248-308)}$ complex at a resolution of 2.15 Å, with two copies of the complex molecule in one asymmetric unit (Supplementary Table 1). In the final model, all residues were visible, except for the residues 38–54 of SNF11$^{(38-169)}$ and the residues 298-308 of SNF2$^{(248-308)}$. Superimposition of the structures of yeast SNF11$^{(38-169)}$/SNF2$^{(248-308)}$ and human BRG1$^{(172-213)}$-SS18$^{(14-101)}$ heterodimers indicated that human SS18/BRG1 and yeast SNF11/SNF2 subcomplexes share similar heterodimer assembling model (Supplementary Fig. 5d). The overall structure of SNF11$^{(38-169)}$/SNF2$^{(248-308)}$ resembles a six-helix bundle, in which SNF2$^{(248-308)}$ is composed of three α-helices [residues 251–267 (αA'), 274–285 (αB'), and 289–296 (αC')] and SNF11$^{(38-169)}$ is also composed of three α-helices [residues 61–94 (α1'), 99–130 (α2'), and 155–168 (α3')] (Fig. 2b and

Supplementary Figs. 2 and 3). The binding of SNF11 to SNF2 was primarily mediated through hydrogen bonds between several pairs of amino acids including, Q63[SNF11] binding to Q252[SNF2], and L70[SNF11], N73[SNF11], or S74[SNF11] binding to Q259[SNF2], individually (Fig. 2c). Moreover, SNF2 residues Q252 and Q259 are conserved from yeast to human (Supplementary Fig. 2). In order to validate the interactions between SNF2 and SNF11 in the heterodimer structure, residues Q252 and Q259, were replaced with alanine. As anticipated, co-IP analysis indicated that the single mutation Q259A weakens the binding of SNF2 to SNF11 dramatically (lane 3 compared to lane 2 in Fig. 2d), and that double mutations Q252A and Q259A completely abolish the interaction between SNF2 and SNF11 (lane 4 compared to lane 2 in Fig. 2d). Collectively, these results confirmed the interaction mode between SNF2 and SNF11 revealed by the structure of the SNF11[(38-169)]/SNF2[(248-308)] heterodimer.

**SS18 undergoes liquid–liquid phase separation in vitro and in vivo.** As previously mentioned, our structural and biochemical studies showed that the bindings of human SS18 or oncofusion SS18-SSX1 to BRG1 and yeast SNF11 to SNF2 share a similar assembly mode. When compared to SNF11, SS18 contains an extra carboxyl region, including the QPGY domain (Figs. 1a and 2a and Supplementary Fig. 3). Interestingly, an IUPred analysis indicated that the carboxyl region of SS18 is intrinsically disordered as well[23] (Fig. 3a). From this data, along with the observations showing that the QPGY domain is rich in tyrosine residues, which are involved in multivalent interactions to drive protein LLPS, we inferred that SS18 might form condensates in vitro and in vivo[18,24]. As expected, purified fluorescent-labeled SS18 protein (Alexa488-SS18) spontaneously formed micrometer-sized droplets in droplet formation buffer. The droplets, in a protein concentration-dependent manner, grew in number and size, a characteristic phenomenon of LLPS (Fig. 3b). Moreover, the formation of condensed droplets was largely suppressed when 5% 1,6-hexanediol was added to the solution, suggesting that hydrophobic interactions are involved in the process. The 1,6-hexanediol is an aliphatic molecule that is reported to disturb hydrophobic interaction-induced phase separation assemblies both in vitro and in vivo[25–27]. Subsequent light microscopy analysis revealed that SS18 spherical droplets also undergo dynamic fusion events, which is indicative of dynamic, liquid-like properties (Fig. 3c). During protein purification, we found that varying protein storage conditions, such as temperature, can cause the SS18 protein solutions to become opalescent. We, therefore, separated the condensed liquid phase from the bulk aqueous solutions by centrifugation and found that LLPS of SS18 can occur at a wide temperature range (4–37 °C assayed)[28]. Lower temperatures can promote the phase transition of SS18 (Fig. 3d, e).

In order to test whether SS18 can undergo LLPS in living cell lines, exogenous GFP-tagged SS18 (GFP-SS18) was transiently expressed in HEK293T cells. Notably, GFP-SS18 showed a characteristic punctate pattern in the nuclei with foci ranging from 0.2 to 1 μm in size, which is consistent with the observations in prior studies[29,30]. In contrast, GFP-tagged SNF11 (GFP-SNF11), which lack intrinsically disordered regions, are diffusely distributed throughout the entire cell (Fig. 3f and Supplementary Fig. 6a). In addition, fluorescence signal recovery after bleaching (FRAP) experiments in HeLa cells showed that roughly 72.8% of the GFP-SS18 molecules in foci exchanged with their counterparts in the surrounding bulk solvent, with an average recovery half time of 6.9 s (Fig. 3g, h). This result indicated that SS18 is condensed in liquid-like phase droplets formed as puncta in cells. Taken together, SS18 formed phase-separated droplets in vitro and dynamic liquid-like condensates in vivo.

**Tyrosine residues are important for SS18 phase separation.** The QPGY domain of SS18 is composed predominantly of glutamine, proline, and glycine, with tyrosine residues occurring at variable intervals and forms homo-oligomers[9]. Considering the emerging role of the multivalent interactions among tyrosine residues in protein LLPS, we mutated 21 tyrosine residues to serine in the intrinsically disordered region of SS18, herein referred to as SS18(Y21S), to disrupt the multimerization of SS18 (Supplementary Fig. 3)[18,24]. As expected, co-IP analysis showed that the mutant SS18(Y21S) loses its ability of self-association (lane 4 compared to lane 2 in Fig. 4a). When compared with the wild-type SS18 (SS18-WT) protein, the sedimentation experiments showed that all the mutant proteins SS18(Y21S) remain in the aqueous supernatant after centrifugation (lanes 5 and 6 compared to lanes 1 and 2 in Fig. 4b, c). Consistent with this observation, purified SS18(Y21S) proteins showed a diffuse distribution in HEK293T cells could not form any droplets (Fig. 4d, e). To further examine the effect of tyrosine residues in phase separation, the QPGY domain of SS18 was fused to the carboxyl-terminal of SNF11, and this chimera was named SNF11-QPGY. Supplementary Fig. 5e, f provide several lines of evidence demonstrating that SNF11-QPGY chimera might undergo phase separation via QPGY domain-mediated self-association as well. Collectively, these results indicated that tyrosine residue-mediated multimerization is important for SS18 phase separation.

**SS18 recruits BRG1 into phase-separated condensates.** As the catalytic subunit of several chromatin remodeling complexes, BRG1 specifically binds to the SNH domain of SS18, as previously seen in Fig. 1. Therefore, we inferred that nuclear SS18 condensates might compartmentalize BRG1. To test this hypothesis, we expressed GFP-SS18 along with mCherry-tagged BRG1 (Cherry-BRG1) in HEK293T cells. As expected, we found that BRG1 is readily recruited within SS18 nuclear condensates when coexpressed with SS18 but demonstrates a diffused distribution in the nucleus when SS18 is absent (Fig. 4f). Interestingly, when coexpressed with the mutant SS18(3M), BRG1 was fully separated from the concentrates (Fig. 4f). These results are consistent with the observations showing that the mutant SS18(3M) cannot bind to BRG1 as previously shown in Fig. 1d but is capable of LLPS (Fig. 4b–e). Moreover, we found that BRG1 diffusely colocalizes with the mutant SS18(Y21S) which contains the inability of LLPS, yet is capable of binding to BRG1 (Fig. 4f). We consistently found that BRG1[(1-282)] is enriched within the SS18-WT condensates or pelleted with SS18-WT, while the mutant SS18(3M) can neither recruit BRG1[(1-282)] into the droplets nor pellet down BRG1[(1-282)] (Fig. 4b, c, g). The mutant SS18(Y21S) was used as a control since it could not form any droplets or pellet down itself and BRG1[(1-282)]. Together, these data suggest that phase separation of SS18 leads to the enrichment of BRG1 into nuclear condensates through the specific interaction between SS18 and BRG1.

**SS18-SSX1 undergoes LLPS and recruits BRG1 into condensates.** As mentioned earlier, the oncofusion protein SS18-SSX1 contains almost the entire QPGY domain of SS18 and retains the ability to stably incorporate into the CBAF complex through the SNH domain binding to BRG1 (Fig. 1 and Supplementary Fig. 4a)[13]. In addition, the repression domain (RD domain) of SSX1 can specifically recruit transcriptional regulators to specific genomic loci[14,31]. An IUPred analysis indicated that the carboxyl region, including domains QPGY and RD, of SS18-SSX1 is intrinsically disordered as well (Supplementary Fig. 6b). Due to these observations, we would like to test whether SS18-SSX1 still retains the ability of phase separation and distinguish it from the SS18, which is widely expressed in normal tissues and

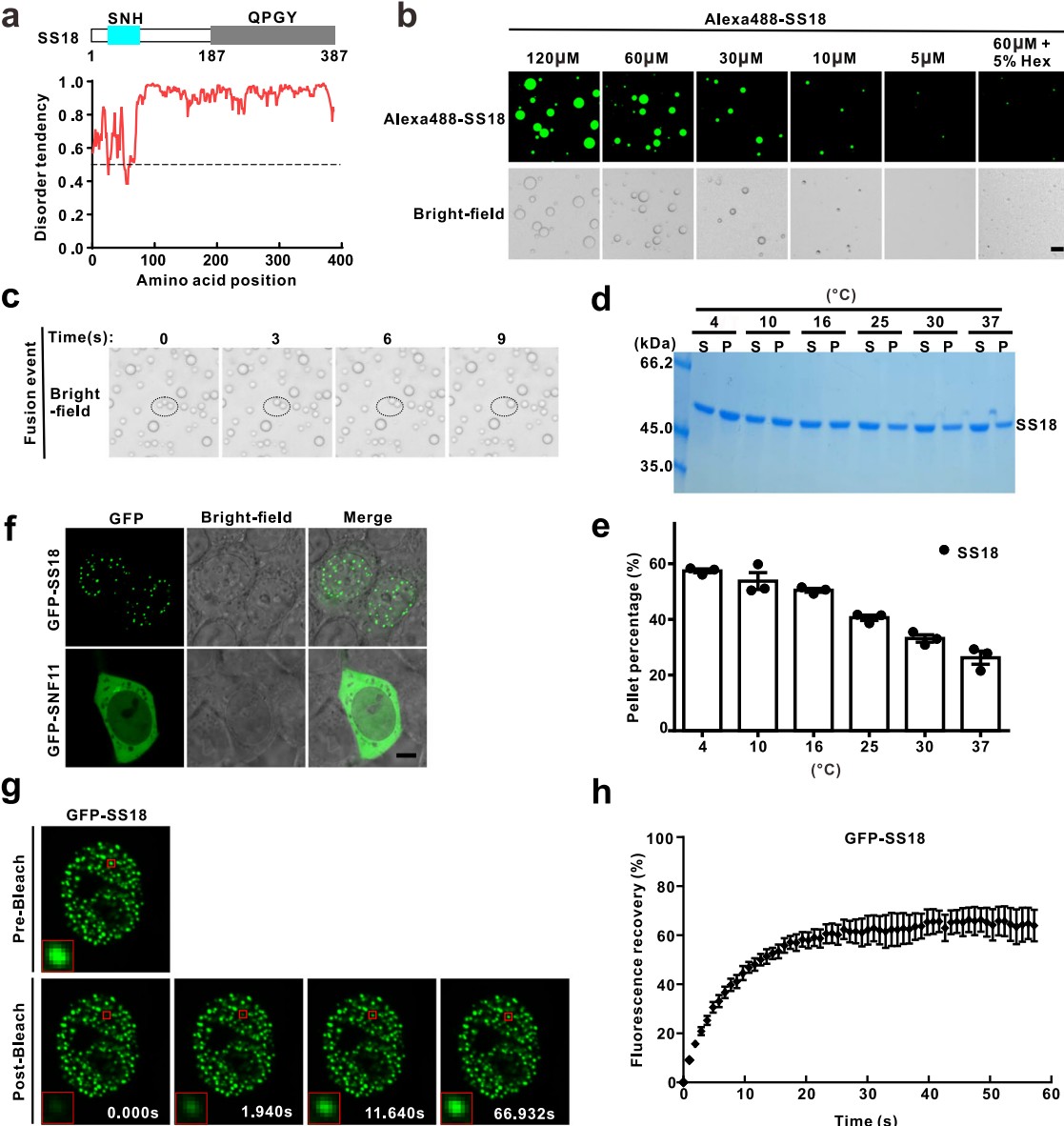

**Fig. 3 SS18 undergoes liquid–liquid phase separation (LLPS) in vitro and in vivo. a** Analysis of SS18 387 amino acid protein sequence. Known domains SNH and QPGY. Red line predicts intrinsically unstructured regions (IUPred) score for intrinsically disordered tendencies; >0.5 is considered disordered. **b** Fluorescence and bright-field images of the SS18 droplets at varying protein concentrations. 1,6-hexanediol (Hex, 5%) was added to the SS18 protein (60 μM) to disrupt droplet formation. Liquid droplets are enriched in Alexa Fluor 488-labeled SS18 (1:100 molar ratio of labeled to unlabeled SS18). This protein labeling ratio was used throughout the study unless otherwise stated. The scale bar indicates 10 μm. **c** The small droplets underwent time-dependent dynamic fusion in the buffer comprised 50 mM Tris-HCl pH 7.5 and 150 mM NaCl at room temperature. Representative SDS-PAGE analysis (**d**) and quantification data **e** showing the distribution of proteins between aqueous solution/supernatant (S) and condensed liquid droplets/pellet (P) fractions for SS18 protein (30 μM) at different temperatures. The band intensities of proteins were quantified with Image J v1.8.0 software. Quantitative data represent results from three independent batches of sedimentation experiments and are plotted as mean ± SEM. **f** Live-cell imaging (GFP) and concurrent phase-contrast imaging for GFP-SS18 and GFP-SNF11 in HEK293T cells. The scale bar indicates 5 μm. **g** Representative time-lapse FRAP images showing that GFP-SS18 signal within the puncta recovered within a few seconds in HeLa cells. Red boxes show the zoomed-in regions. **h** FRAP recovery curves for six GFP-SS18 puncta from independent six HeLa cells with error bars indicating mean ± SEM. Time 0 refers to the time point of the photobleaching pulse.

cells. First, we found that the purified SS18-SSX1 fusion proteins do not form condensed droplets in the same solution compositions and protein concentration as that of SS18. Intriguingly, we noted that the SS18-SSX1 solution turns turbid, and that the droplet formation occurs in a concentration-dependent manner, when buffer contained 10% Ficoll mimicking the crowded environment of the nucleus (Fig. 5a). In addition, small droplets gradually coalesce into larger ones within the first 15 s (Fig. 5b).

We also observed that roughly 40% of SS18-SSX1 proteins were recovered from the condensed phase (pellet fractions) in the presence of 10% Ficoll (lanes 7 and 8 in Fig. 5c, d). In combination with the 10% Ficoll, 1,6-hexanediol (5% Hex) leads to the dispersion of the condensed phase (lanes 13 and 14 compared to lanes 7 and 8 in Fig. 5c, d). However, we found that purified SSX1[(111-188)] itself cannot undergoes LLPS under the same conditions as SS18-SSX1 protein (Supplementary Fig. 6c). Together,

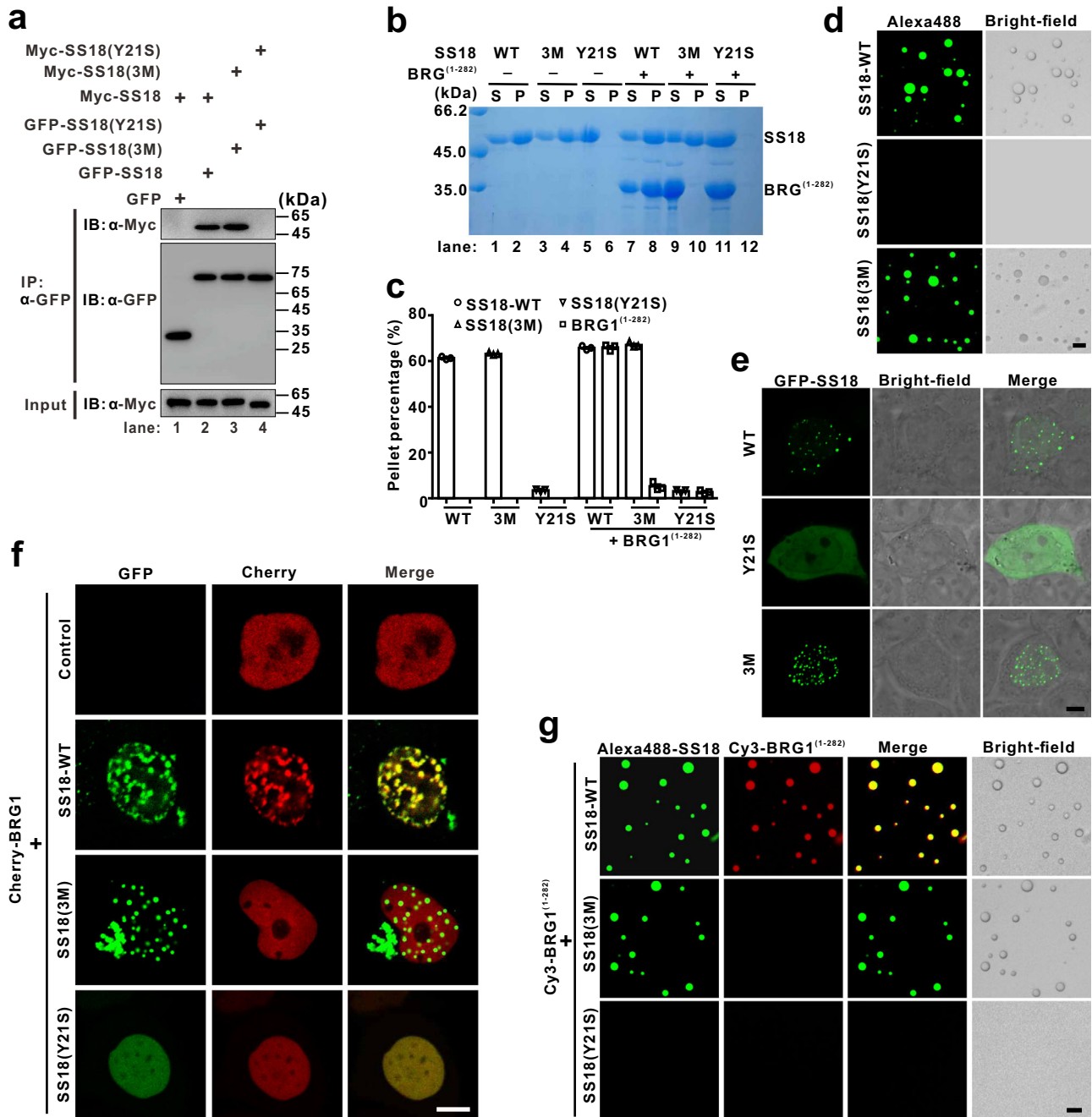

**Fig. 4 Specific recruitment of BRG1 into SS18 phase-separated condensates through tyrosine residue-mediated multimerization. a** Co-IP experiments testing the self-association ability of SS18 or its mutants. Extracts were prepared from HEK293T cells transfected with combinations of plasmids, as indicated. The bottom panel shows 3% of the Myc-SS18 as input for each IP. Representative SDS-PAGE analysis (**b**) and quantitative data (**c**) for sedimentation assay of SS18 WT, mutants, or various SS18/BRG1$^{(1-282)}$ mixtures, as indicated. The concentration of each protein is 60 μM. The band intensities of proteins were quantified with Image J v1.8.0 software. The statistical data represent results from three independent batches of sedimentation experiments and are plotted as mean ± SEM. **d** Fluorescence and bright-field images of SS18 WT or mutant droplets at the protein concentration of 60 μM. Liquid droplets are enriched in Alexa Fluor 488-labeled SS18 WT or mutant. The scale bar indicates 10 μm. **e** Representative fluorescence images of GFP-SS18(WT), GFP-SS18(Y21S), and GFP-SS18(3M) in HEK293T cells. The scale bar indicates 5 μm. **f** Co-localization of GFP-SS18-WT or mutants and Cherry-BRG1 in HEK293T cells. The scale bar indicates 5 μm. **g** Representative fluorescence and bright-field images of the mixture of Alexa Fluor 488-labeled SS18 or mutants (60 μM) and Cy3-labeled BRG1$^{(1-282)}$ (60 μM) in the buffer comprised 50 mM Tris-HCl pH 7.5 and 150 mM NaCl at room temperature. The scale bar indicates 10 μm.

these results suggested that the oncofusion protein SS18-SSX1 undergoes LLPS in vitro in the presence of crowding reagents.

Next, we sought to test the ability of phase separation of two mutants SS18(3M)-SSX1 and SS18(Y19S)-SSX1, in which containing three mutations (I32E, L54E, and A65E) and 19 tyrosine

residues mutated to serine in SS18, respectively (Fig. 1a and Supplementary Fig. 3). As anticipated, a similar amount of the mutant SS18(3M)-SSX1 and SS18-SSX1-WT proteins were recovered from the condensed phase (lanes 9 and 10 compared to lanes 7 and 8 in Fig. 5c, d). These results are congruous with

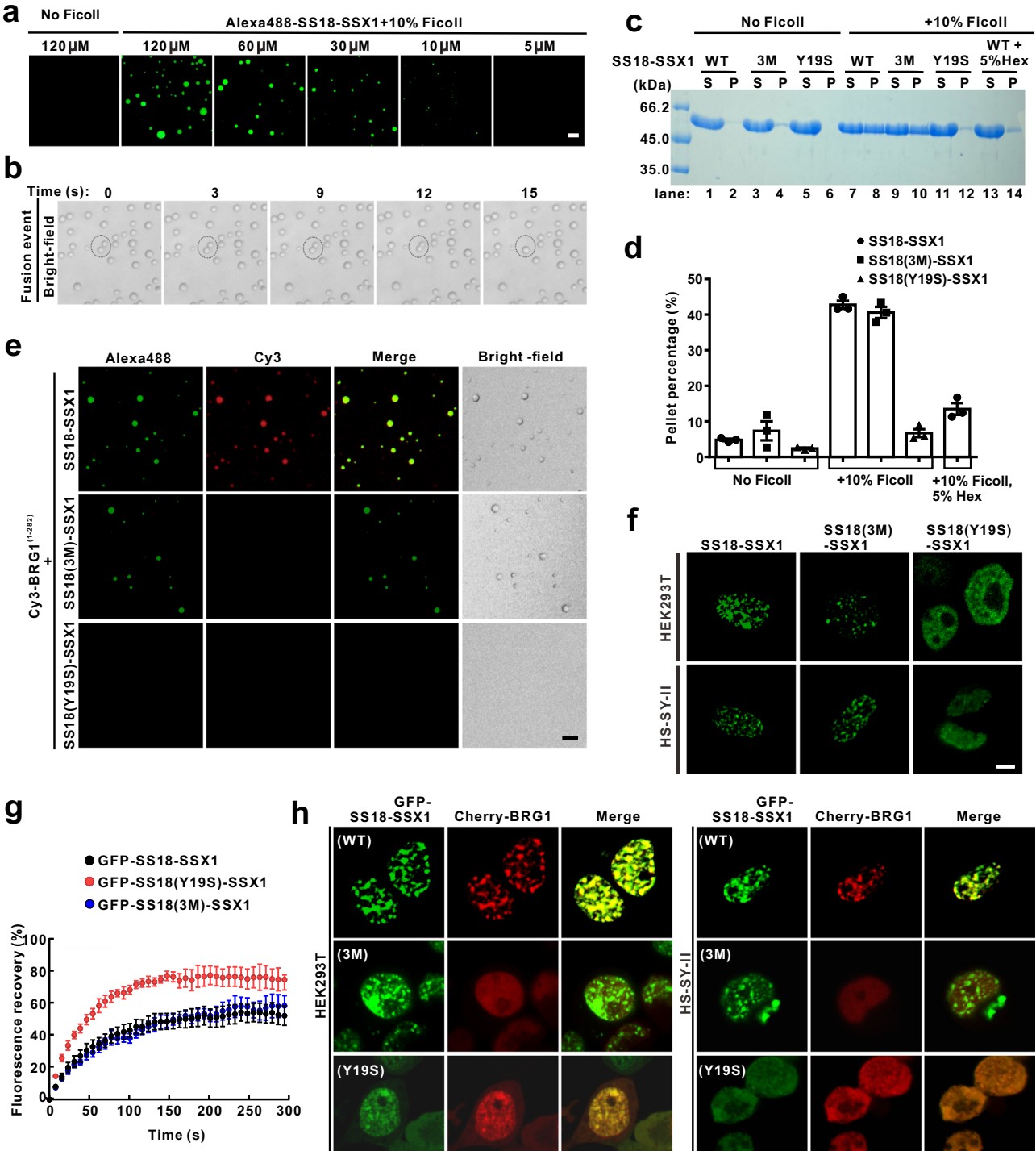

**Fig. 5 SS18-SSX1 undergoes LLPS and recruits BRG1 into condensates. a** Representative fluorescence images of the fusion protein SS18-SSX1 at different protein concentrations. Liquid droplets are enriched in Alexa Fluor 488-labeled SS18-SSX1. The droplet formation buffer comprises 50 mM Tris-HCl pH 7.5, 150 mM NaCl, and 10% Ficoll. The scale bars indicate 10 μm. **b** Phase-contrast imaging of SS18-SSX1 protein at different time points to show the dynamic fusion of two droplets. Representative SDS-PAGE analysis (**c**) and quantitative data (**d**) of the sedimentation assay of SS18-SSX1 protein or mutants in the buffer with or without 10% Ficoll. 1,6-hexanediol (Hex, 5%) was added to the SS18-SSX1 protein to disrupt droplet formation. The concentration of each protein is 60 μM. The band intensities of proteins were quantified with Image J v1.8.0 software. The statistical data represent results from three independent batches of sedimentation experiments and were plotted as mean ± SEM. **e** Representative fluorescence and bright-field images of the mixture of Alexa Fluor 488-labeled SS18-SSX1 or mutants (60 μM) and Cy3-labeled BRG1$^{(1-282)}$ (60 μM) in the buffer comprised 50 mM Tris-HCl pH 7.5, 150 mM NaCl, and 10% Ficoll at room temperature. The scale bar indicates 10 μm. **f** Live-cell imaging for GFP-SS18-SSX1, GFP-SS18(3M)-SSX1, and GFP-SS18(Y19S)-SSX1 in HEK293T and HS-SY-II cells. The scale bar indicates 5 μm. **g** FRAP recovery curves for GFP-SS18-SSX1, GFP-SS18(3M)-SSX1, and GFP-SS18(Y19S)-SSX1 puncta from three independent HeLa cells with error bars indicating mean ± SEM. Time 0 refers to the time point of the photobleaching pulse. **h** Co-expression of GFP-SS18-SSX1 or mutants and Cherry-BRG1 in HEK293T and HS-SY-II cells. The scale bar indicates 5 μm.

earlier co-IP analyses in Supplementary Fig. 4b which indicates that the mutant SS18(3M)-SSX1 and SS18-SSX1-WT are capable of self-association (Supplementary Fig. 4b). In addition, the mutant SS18(Y19S)-SSX1 did not recover from the condensed phase, further proving that tyrosine residues-mediated multivalent interactions of the QPGY domain of SS18 are the driving force for the LLPS of SS18-SSX1 (lanes 11 and 12 in Fig. 5c, d). It is also important to note that only the wild-type SS18-SSX1, is able to recruit BRG1$^{(1-282)}$ into the condensed droplets (Fig. 5e). Despite having the ability to bind to BRG1, the mutant SS18(Y19S)-SSX1 could not form any condensed droplets, whereas the mutant SS18(3M)-SSX1 was able to form condensed droplets, but not able to recruit BRG1 (Fig. 5e). Subsequently, we observed that both SS18-SSX1 and the mutant SS18(3M)-SSX1 show a more dense puncta-like cluster, distributed in cells HEK293T and synovial sarcoma cell-line HS-SY-II or CME-1, when compared to SS18 puncta previously shown in Fig. 4e (Fig. 5f and Supplementary Fig. 4c). In contrast, the mutant SS18(Y19S)-SSX1 diffused in these cells and a small amount of protein aggregation was observed (Fig. 5f and Supplementary Fig. 4c). To detail the properties of the puncta cluster, we used FRAP to monitor the exchange rate of SS18-SSX1 or two mutants molecules in the puncta cluster with their counterparts in the surrounding bulk solvent. After bleaching, about 54–58% of GFP-SS18-SSX1 and GFP-SS18(3M)-SSX1 puncta signal were recovered within an average half-life time of 38.7 s and 49.8 s, respectively, which are slower than that of SS18 (Figs. 3h and 5g). Notably, roughly 76% of the GFP-SS18(Y19S)-SSX1 fluorescence in the cluster recovered in an average half-life time of 28.2 s, indicating that the mutations of tyrosine to serine in the QPGY domain lead to more dynamic characteristics of this mutant (Fig. 5g). Mutant SS18(3M)-SSX1 proved its incapability of colocalizing with BRG1, whereas SS18-SSX1-WT was able to recruit BRG1 into the phase-separated condensates in cells HEK293T and synovial sarcoma cell-line HS-SY-II or CME-1 (Fig. 5h and Supplementary Fig. 4d). In addition, mutant SS18(Y19S)-SSX1, which has the inability of phase separation, was observed to have a diffused distribution of BRG1 in the cell lines as well (Fig. 5h and Supplementary Fig. 4d). Taken together, these results indicated that the oncofusion SS18-SSX1 recruits BRG1 into phase-separated condensates through specific interactions between SS18-SSX1 and BRG1.

**LLPS and binding BRG1 benefit SS18-SSX1's transformation activity.** Next, we wanted to explore the role of SS18-SSX1 phase separation and BRG1 binding on SyS development. To address this, we reconstituted three NIH3T3 fibroblast cell lines that stably overexpress SS18-SSX1, SS18(3M)-SSX1, and SS18(Y19S)-SSX1. Western blot results showed that the expression levels of SS18-SSX1, SS18(3M)-SSX1, and SS18(Y19S)-SSX1 in each cell line were comparable and were significantly increased compared with the control (Fig. 6a). In order to test the role of SS18-SSX1 on tumorigenesis, we performed EdU cell proliferation and clonogenic assays to detect the synthesis of DNA and clone-formation abilities. Notably, EdU proliferation experiments demonstrated that when compared to the control, overexpression of WT SS18-SSX1 efficiently promotes NIH3T3 cells proliferation, whereas both mutants SS18(3M)-SSX1 and SS18(Y19S)-SSX1 exhibit partial deficiency in promoting cell proliferation (Fig. 6b, c). Colony-formation experiments determined that NIH3T3 cells expressing either mutants SS18(3M)-SSX1 or mutant SS18(Y19S)-SSX1 form few and smaller clones compared to wild-type (Fig. 6d). In addition, we tested the migration and invasion abilities of NIH3T3 cells stably expressed WT SS18-SSX1 as well as mutants. These results showed that WT SS18-

SSX1, but not the mutants, promote both migration and invasion of NIH3T3 cells (Fig. 6e–h). Taken together, these results indicated that phase separation of the oncoprotein SS18-SSX1 and its assembly into chromatin remodeling complexes are important for its transforming activity in fibroblast cell NIH3T3.

## Discussion

In this study, we found that human SS18 binds to BRG1 and yeast SNF11 binds to SNF2, the yeast homolog of BRG1, with a similar heterodimer structure (Figs. 1b and 2b and Supplementary Fig. 5d). This indicates the conservation between SNF11 and the N-terminal SNH domain of SS18. With this observation and the sequence similarities of SNF11 and the SNH domain of SS18 across a broad range of species, we are able to conclude that SNF11 might be a homolog of SS18 in *S. cerevisiae* (Supplementary Fig. 3). SS18 has an extra intrinsically disordered region, including the QPGY domain, that mediates LLPS through multivalent hydrophobic interactions, while its yeast homolog, SNF11, does not have the ability to phase separation (Fig. 3). Moreover, the chimera SNF11-QPGY also demonstrated to undergo phase separation (Supplementary Fig. 5e, f). With this in consideration, we speculated that SS18 acquired a characteristic of phase separation to participate in specific physiological functions in *Chordata* in the process of evolution. Interestingly, a recent study showed that SS18 mediates CBAF assembly through phase separation to regulate pluripotent-somatic transition, consistent with our speculation[32]. In their study, the authors found that a C-terminal intrinsically disordered region of SS18 regulates its forming microscopic condensates under the condition of ectopic overexpression, and that the N-terminal 70aa of SS18 is required for its binding to CBAF. These observations are consistent with our results. Moreover, we used purified recombinant proteins to confirm SS18 LLPS in vitro and revealed the structural basis of the interaction between SS18 and BRG1.

We evaluated phase separation of SS18 or SS18-SSX1 in live cells using ectopic overexpression systems (Figs. 3–5). In addition, we noted that the endogenous SS18 or SS18-SSX1 shows as a similar puncta-like distribution in mouse embryonic stem or synovial sarcoma cells as that of overexpression in HeLa, HEK293T, or synovial sarcoma cells[30,32] (Figs. 3–5). It is an undeniable that, however, LLPS is a concentration-dependent process. Although it is difficult to assess LLPS in native state systems[33], future studies are needed to validate SS18 or SS18-SSX1 LLPS in an appropriate in vivo system.

Our results showed that the characteristics of droplets/condensates formed by SS18 and SS18-SSX1 are slightly different, although the driving force to undergo LLPS is similar (Figs. 3–5). Specifically, SS18-SSX1 compared to SS18 can form phase-separated droplets only in the presence of a certain concentration of crowding reagent. In addition, SS18-SSX1 fluorescent molecules are shown to exchange at a much slower rate than SS18 with their counterparts in bulk solvent (Figs. 3 and 5). These results suggested that the fusion partner SSX1$^{(111-188)}$ might contribute to or regulate LLPS of the oncofusion SS18-SSX1. It has also been reported that SSX proteins are diffusely distributed in the nuclei of SyS or fibrosarcoma cells, along with some nuclear dots[34,35]. Consistently, our results showed that purified SSX1$^{(111-188)}$ itself cannot occur in phase separation in vitro (Supplementary Fig. 6c). Due to this, the exact mechanism of the SSX1$^{(111-188)}$ in regulating phase separation of SS18-SSX1 and its underlying biological significance still needs to be further studied.

The dysregulation of chromatin-based gene-regulatory systems is thought to be a central driver of SyS pathogenesis[36]. A prior study pointed to an interesting ability of SS18-SSX, which recruits polycomb repressive complex 2 (PRC2) to ATF2 target genes to

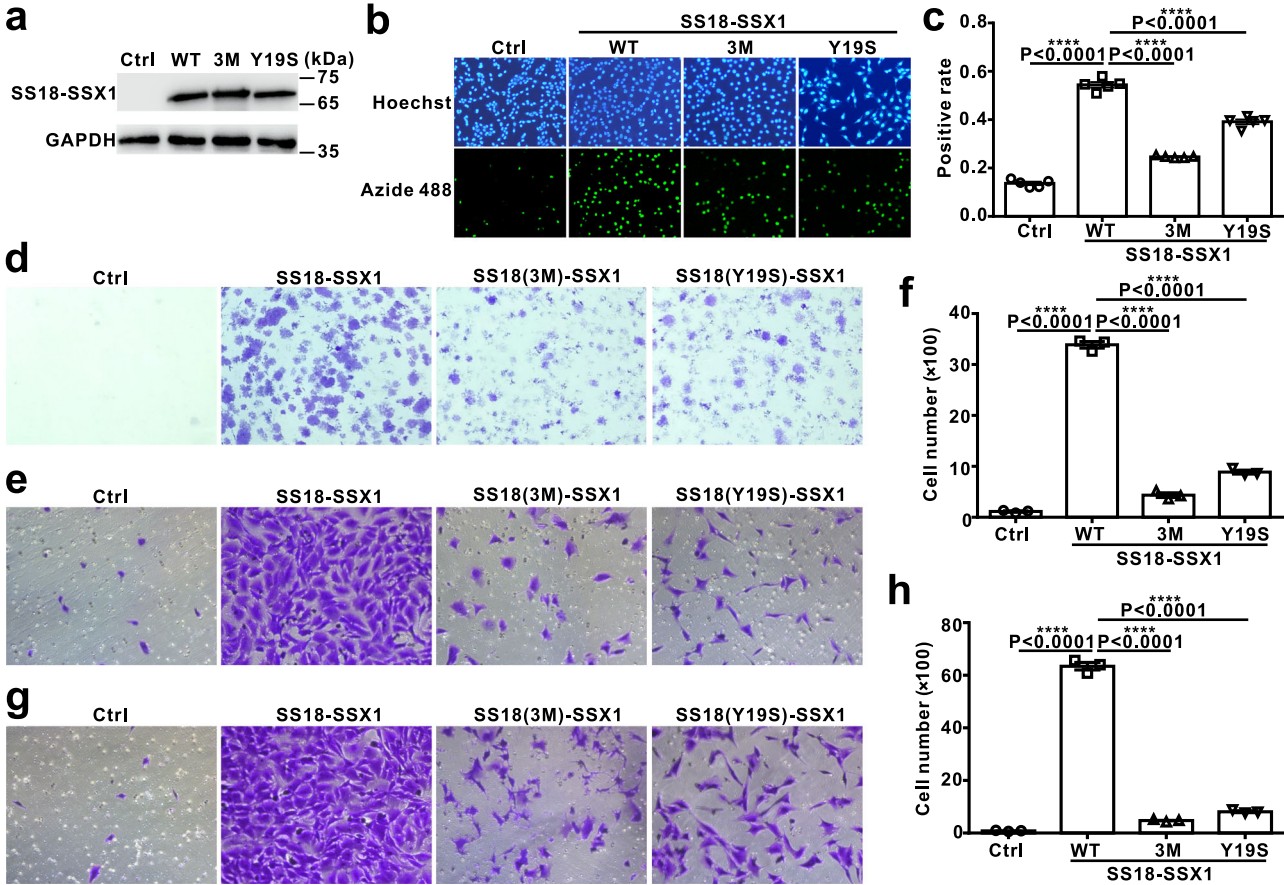

**Fig. 6 Phase separation of SS18-SSX1 and binding to BRG1 are important for the oncogenic activity of SS18-SSX1. a** The protein level of SS18-SSX1 WT or mutants overexpressed in NIH3T3 cells. **b** EdU assay of NIH3T3 cell lines overexpressed SS18-SSX1 WT or mutants. EdU positive nuclei (Alexa Fluor 488 azide labeled; green) and 1× Hoechst stained nuclei of all the cells (blue) were visualized by fluorescence microscopy. **c** Statistical graph of the EdU assay in (**b**). Error bars represent mean ± SEM, one-way ANOVA, Tukey's multiple comparisons test; $n = 5$ independent experiments, ****$P < 0.0001$. **d** Plate colony formation of NIH3T3 cell lines overexpressing SS18-SSX1 WT and mutants. Colonies were stained with 0.1% crystal violet. Cell migration (**e**) and invasion **g** assay of NIH3T3 cell lines overexpressing SS18-SSX1 WT and mutants. Cells were stained with 1% crystal violet and visualized by bright-field microscopy. **f**, **h** Statistical graph of the cell migration and invasion assay. Error bars represent mean ± SEM, one-way ANOVA, Tukey's multiple comparisons test; $n = 3$ independent experiments, ****$P < 0.0001$.

repress tumor-suppressor genes encoded by the CDKN2A locus[37]. On the other hand, other studies suggest that SS18-SSX alters BAF genome distribution and subtype levels and therefore increases the complex's ability to disrupt PRC2 function. This disruption could mediate activation of gene expression through the COMPASS family of histone H3K4 methylases and orchestrate aberrant oncogenic transcriptional programs[13,38,39]. A study from Banito et al. found that SS18-SSX fusions associate with KDM2B-PRC1.1, a noncanonical polycomb repressive complex 1, to aberrantly activate transcription factors that are targets of polycomb-mediated gene repression[40]. Subsequent research elucidated that the SS18-SSX-specific conformation of BAF complexes exhibit a strong preference for H2AUbK119-marked nucleosomes, supporting their preference for polycomb-decorated chromatin regions[31]. These observations, together with our results, suggest that phase separation is an important mechanism. This mechanism enables SS18-SSX1 condensates to function as hubs in compartmentalizing chromatin remodeling complex(es) in an efficient and specific manner that drives synovial sarcomagenesis. Future studies are needed to further detail the effect and role of SS18-SSX LLPS in an animal model.

Targeting LLPS-related mechanisms to treat a range of diseases is a promising idea[41]. For example, a small-molecule inhibitor of nuclear PARP-1/2 activity was developed to treat amyotrophic

lateral sclerosis (ALS) by reducing the formation of cytoplasmic TDP-43 aggregates in mammalian cells[42]. Considering our results and these previous studies small molecules targeting the phase separation of SS18-SSX1 could be an attractive form of therapy for SyS.

## Methods

**Protein expression and purification.** Human *SS18* (residues 1–387, *SS18*[(1-387)]), yeast *SNF11* (residues 1–169, *SNF11*[(1-169)]), and yeast *SNF2* (residues 248–308, *SNF2*[(248-308)]) were amplified by PCR from the cDNA library of the HEK293T cell line and the *Saccharomyces cerevisiae* genome DNA, respectively. Human *BRG1* was purchased from Addgene (Plasmid #1959). *SSX1* and *SS18(Y21S)* were obtained through gene synthesis (GENEWIZ). Mutations were generated using a standard PCR-based mutagenesis method and confirmed by DNA sequencing. To make a single-chain fusion protein of the BRG1[(172–213)] and SS18[(14–101)], DNA fragments were amplified by PCR and linked with a TEV protease-cleavable segment (Glu–Asn–Leu–Tyr–Phe–Gln–Ser). Two amino acids (Ser–Gly) were inserted on both sides of the TEV segment. Then, the single-chain was cloned into an in-house modified version of the pET32a vector (Novagen, 69015-3CN) and the resulting protein contained a thioredoxin (Trx)-his6 tag on its N-terminus. *SNF11*[(38-169)] and *SNF2*[(248-308)] were cloned into two multiple cloning sites of pETDuet-1 vector (Novagen, 71146-3) separately and sequentially. For proteins studied in phase separation: *SS18-SSX1*, *SS18(3M)-SSX1*, *SS18(Y19S)-SSX1*, and *BRG1*[(1-282)] were cloned into an in-house modified version of the pET32a vector. The resulting proteins contained a Trx-his6 tag on the N-terminus. *SS18*, *SS18(3M)*, and *SS18(Y21S)* were cloned into an in-house modified version of the pET32a vector with a maltose-binding protein (MBP)-his6 tag on the N-terminus (Supplementary Table 2).

BL21(DE3) Codon Plus *Escherichia coli* cells harboring the expression plasmid were grown in LB medium at 37 °C until the $OD_{600}$ reached 0.6, and protein expression was induced with 300 μM isopropyl-β-D-thiogalactoside (Chemsynlab, A04283) at 16 °C for 16–18 h. The Se-Met-substituted protein was expressed in methionine auxotrophic *E. coli* B834 (DE3) cells grown in LeMaster medium. Proteins were purified by $Ni^{2+}$-NTA agarose affinity chromatography followed by size-exclusion chromatography on a HiLoad 26/60 Superdex 200 (GE Healthcare) in 50 mM MES pH 6.0, 100 mM NaCl, 1 mM EDTA, and 1 mM DTT. After digestion with PreScission Protease to cleave the N-terminal Trx-his6 tag, the target protein was further purified on a HiTrap SP HP anion-exchange column. The final purification step was size-exclusion chromatography on a Superdex 200 10/300 increase column (GE Healthcare) in 50 mM MES pH 6.0, 100 mM NaCl, and 1 mM DTT.

**Analytical ultracentrifugation.** SV experiments were performed in a Beckman Coulter XL-I analytical ultracentrifuge (Beckman Coulter) using double sector centerpieces and sapphirine windows. The proteins were changed into the buffer containing 50 mM MES pH 6.0, 50 mM NaCl, 1 mM EDTA, and 1 mM DTT by a Superdex 200 10/300 increase column before the experiments. SV experiments were conducted at 4 °C using interference light detection. The SV data were analyzed using the SEDFIT v14.0 program[43,44].

**Crystallization and data collection.** Both native and Se-Met-substituted crystals of human $BRG1^{(172-213)}$-$SS18^{(14-101)}$ complex were obtained by the sitting drop vapor diffusion method at 20 °C. Crystals were grown in the solution containing 1.16 M sodium phosphate monobasic monohydrate and 0.24 M potassium phosphate dibasic at a protein concentration of 40 mg/mL. Native crystals of yeast $SNF11^{(38-169)}$/$SNF2^{(248-308)}$ complex were grown at 20 °C at a protein concentration of 7 mg/mL using the same method as above. The protein was equilibrated against a reservoir solution of 0.79 M sodium phosphate monobasic monohydrate and 0.61 M potassium phosphate dibasic. To obtain phase information, the crystals of $SNF11^{(38-169)}$/$SNF2^{(248-308)}$ were soaked in 2 mM Mercury (II) acetate (Hampton, HR2-446) for 30 min. All crystals were cryo-cooled in the precipitant solution containing 25% (v/v) glycerol using liquid nitrogen and all diffraction data were collected at the Shanghai Synchrotron Radiation Facility (SSRF) on beamlines BL19U1[45] and were processed using the HKL2000 v714 software package[46].

**Structure determination and refinement.** Phasing and initial model building of human $BRG1^{(172-213)}$-$SS18^{(14-101)}$ complex crystal structure was determined by single-wavelength anomalous dispersion (SAD) using PHENIX v1.15-2155 AutoSol wizard[47] and AutoBuild wizard[48], respectively. The initial phases and models of yeast $SNF11^{(38-169)}$/$SNF2^{(248-308)}$ complex were determined by SAD using the Shelx C/D/E program[49] in CCP4i v7.0.073. Then, the initial models were further rebuilt and adjusted manually with the Coot v0.8.3 program[50] and were refined by Phenix v1.15-2155 refinement program (https://www.phenix-online.org/). The final model was further validated using MolProbity[51]. Detailed data collection and refinement statistics are summarized in Supplementary Table 1. All structural figures were prepared using PyMOL v1.6 (http://www.pymol.org/) and the planar graph of protein-protein interaction were generated by Ligplot+ v2.2 software[52].

**Cell culture and transfection.** HEK293T cells (CBTCCCAS, GNHu17), HeLa cells (CBTCCCAS, TCHu187), NIH3T3 cells (CBTCCCAS, SCSP-515), and HS-SY-II cells (from Hiroshi Sonobe, Department of Pathology, Kochi Medical School, Nankoku, Japan) were maintained in Dulbecco's modified Eagle's medium (Sigma-Aldrich, D6429) supplemented with 10% fetal bovine serum (Biological Industries, 04-010-1 A), 100 U/mL penicillin and 100 μg/mL streptomycin (Hyclone, SV30010) stored in an incubator containing 5% $CO_2$ at 37 °C. CME-1 cells (from Nicolo Riggi, Division of Experimental Pathology, Institute of Pathology, Centre Hospitalier Universitaire Vaudois and University of Lausanne, Lausanne, Switzerland) were cultured in RPM1 1640 containing 10% fetal bovine serum and penicillin–streptomycin. The plasmids were transfected by polyethyleneimine (Polysciences, 02371-500) according to the manufacturer's protocol. For plasmids involved in stable overexpression cell lines, the wild-type *SS18-SSX1* and mutant genes were subcloned into the expression vector pSIN-EF2-Pur (Addgene, Plasmid #16579) with an N-terminal Myc-tag, named as pSIN-EF2-*SS18-SSX1*, pSIN-EF2-*SS18(3M)-SSX1*, and pSIN-EF2-*SS18(Y19S)-SSX1* (Supplementary Table 2). To prepare lentiviral particles, recombinant vector along with pSPAX2 packaging vector and the virus-expressing envelope vector PMD2.G were transfected into HEK293T cells. NIH3T3 cells were seeded in 6-well plates and infected with different viruses on the following day, and 10 μg/mL polybrene (Sigma-Aldrich, H9268) was added to increase the infection efficiency. After 48 h infection, a fresh medium containing 1.5 μg/mL puromycin (Solarbio, P8230) was used to screen positive clones. Positive clones were selected to establish the cell lines' stably expression of empty vector, SS18-SSX1, SS18(3M)-SSX1, or SS18(Y19S)-SSX1. The loading control proteins in each cell line were examined by western blotting with HRP-conjugated GAPDH antibody (Proteintech, HRP-60004) with a dilution of 1:10,000.

**Co-immunoprecipitation.** HEK293T cells were transfected with the indicated combinations of plasmids. After 24 h transfection, HEK293T cells were lysed using ice-cold cell lysis buffer (50 mM Tris-HCl pH 7.4, 150 mM NaCl, 8% glycerol, 0.5% NP40, 0.5% Triton X-100, 1 mM phenylmethylsulfonyl fluoride, and protease inhibitor cocktails) and cleared by centrifugation at 11,750×*g* for 20 min at 4 °C. The supernatants were then incubated with agarose conjugated anti-GFP antibodies for 30 min at 4 °C. The agarose beads were washed three times with cell lysis buffer and eluted with SDS sample buffer. Samples were then subjected to SDS-PAGE and western blot analysis.

**Western blotting.** Proteins were separated by SDS-PAGE and transferred to polyvinylidene difluoride (PVDF) membrane (Millipore, IPVH00010). The membranes were subsequently blocked with 10% nonfat milk in TBST (50 mM Tris-HCl pH 7.4, 150 mM NaCl, and 0.1% Tween 20) for 1 h. The PVDF membranes were immunoblotted with anti-Myc antibody (Sigma-Aldrich, M4439) diluted 1:5000 and anti-GFP antibody (Sigma-Aldrich, G1544) diluted 1:5000 at room temperature for 1 h, and then probed with Goat anti-mouse IgG-HRP (Santa Cruz, sc-2005) or Goat anti-rabbit IgG-HRP (Santa cruz, sc-2004) with a dilution of 1:5000, respectively, and developed with a chemiluminescent substrate (Millipore, WBKLS0500). Protein bands were visualized on the Tanon-5200 chemiluminescent imaging system (Tanon Science and Technology).

**In vitro phase separation assay.** Purified SS18 and $BRG1^{(1-282)}$ protein were prepared in buffer containing 50 mM Tris-HCl pH 7.5 and 150 mM NaCl. SS18-SSX1 protein was diluted to the desired concentration in the droplet formation buffer (50 mM Tris-HCl pH 7.5, 150 mM NaCl, and 10% Ficoll). Formation of phase separation was assayed either directly by imaging-based methods or by sedimentation experiment[28]. Observation and characterization of droplets were carried out on a fluorescence microscope (LEICA CTR5000), and the data were collected by the Leica Application Suite v4.4.0 software. For protein fluorescent labeling, Alexa488-NHS ester (Yeasen, 40779ES03) and Cy3-NHS ester (Yeasen, 40777ES03) were incubated with SS18 or SS18-SSX1 and $BRG1^{(1-282)}$, respectively, at room temperature for 1 h. The fluorophore to protein molar ratio was 2:1 and the solution pH was adjusted to pH 8.3 by 100 mM $NaHCO_3$. The reaction was quenched by 200 mM Tris-HCl pH 8.3. The labeled proteins were further changed into the buffer (50 mM Tris-HCl pH 7.4 and 150 mM NaCl) by the Hitrap desalting column (GE Healthcare).

**Confocal imaging.** HEK293T, HS-SY-II, and CME-1 cells were cultured in glass-bottom dishes (Nest, 801002) and transfected with indicated plasmids as described above. After 24 h transfection, cells were imaged using a Zeiss LSM710 confocal microscope by a ×63 oil-immersion lens, and then the data were collected and processed by the Zen Black v2011 software. For the FRAP assay, HeLa cells were also cultured in glass-bottom dishes and transfected with indicated plasmids as described above. The FRAP assay was also performed on a Zeiss LSM710 confocal microscope at 37 °C. The fluorescence signal of GFP was bleached using a 488-nm laser beam. The fluorescence intensity difference between pre-bleaching and at time 0 (the time point right after the photobleaching pulse) was normalized to 100%.

**EdU cell proliferation assay.** The proliferation of cells was detected using EdU cell proliferation assay according to the manufacturer's instructions. About $1 \times 10^5$ cells were seeded in 12-well plates and maintained for 24 h before the assay. A total of 500 μL EdU (10 μM) reagent (Beyotime, C0071S) was added to each well and incubated for 2 h to label the cells. After three times wash with PBS, cells were fixed in a 4% paraformaldehyde solution (Dingguo Biotechnology, AR-0211) for 15 min, permeabilized with 0.3% Triton X-100 (GenStar, VA11410) for another 15 min, and then incubated with the click-reaction reagent for 30 min at room temperature in the dark environment. In all, 1× Hoechst33342 reagent was used to counterstain the nucleus. The result of staining was observed with a fluorescence microscope system Nikon ECLIPSE Ti-S, and the data were collected by the NIS-Elements F v4.0 software.

**Plate clone-formation assay.** The NIH3T3 cells stably expressed empty vector or wild-type SS18-SSX1, and its mutants were plated in six-well plates at $1.0 \times 10^3$ cells per well in growth medium supplemented with 10% FBS and incubated at 37 °C for 3 weeks. The culture medium was changed every 3–5 days. After 3 weeks, cells were fixed in 4% paraformaldehyde (Dingguo Biotechnology, AR-0211) for 30 min at room temperature and stained with 0.1% crystal violet (Sigma-Aldrich, V5265) for 30 min at room temperature, then cells were washed with water and finally took a picture after air-dried. These experiments were conducted in triplicates.

**Cell migration and invasion assays.** Cell migration and invasion assays were carried out using a 24-well transwell chamber system (Corning, 3422). A transwell apparatus was separated into upper and lower compartments by polycarbonate filters (8-μm pores). For migration assays, the NIH3T3 cells stably expressed empty vector or wild-type SS18-SSX1, and its mutants ($4.0 \times 10^4$) were suspended in

200 μL of growth medium in the upper chamber. For invasion assays, the polycarbonate filters were coated with 300 μg/mL matrigel (Corning, 356234) before the cell seeding. The lower chamber contained a 750 μL growth medium supplemented with 10% FBS. After 24 h incubation at 37 °C in a 5% $CO_2$ incubator, cells on the upper filter surface were removed by wiping with a cotton swab. Filters were then fixed in 4% paraformaldehyde (Dingguo Biotechnology, AR-0211) and stained with 0.1% crystal violet (Sigma-Aldrich, V5265). All cells that migrated to the lower filter surface were counted under a microscope at ×200 magnification. Each assay was performed in triplicates.

**Statistics and reproducibility**. Each experiment of analytical gel filtration and SDS-PAGE (Supplementary Figs. 1a–e and 5a, b), Co-IP assays (Supplementary Figs. 1i, 4a, b, and 5f), and western blot assays (Fig. 6a) were performed twice independently with similar results and one representative result was shown. For the data of phase separation, droplet formation assays in vitro (Figs. 3b, 4d, g, and 5a, e) and confocal images of living cells (Figs. 3f, 4e, f, and 5f, h) were acquired from three independent experiment, and more than 6 images were taken for each sample. They showed similar results, so the representative microscopy images were shown. The sedimentation experiment (Supplementary Fig. 6c) was repeated three times with similar results. Statistical analysis was performed using GraphPad Prism v8.0 software. The data are presented as mean ± SEM as indicated in the figure legends. One-way ANOVA, Tukey's multiple comparisons test, **** for $P < 0.0001$.

**Reporting summary**. Further information on research design is available in the Nature Research Reporting Summary linked to this article.

## Data availability

The nucleotide sequence of yeast SNF11 and SNF2 were obtained from Saccharomyces cerevisiae genome databases with SGD ID of S00002480 and S00005816, respectively. The atomic coordinates and structure factors data for the crystal structure of the BRG1/SS18 and SNF11/SNF2 complex have been deposited in the Protein Data Bank database under accession codes 7VRB and 7VRC, respectively. Source data are provided with this paper. The authors declare that all data supporting the finding of this study are available within this article and its Supplementary Information files.

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

## Acknowledgements

We thank the staffs from BL19U1 beamline of the National Facility for Protein Science in Shanghai (NFPS) at Shanghai Synchrotron Radiation Facility for assistance during data collection and we thank for Wenxin Long for the English editing. This work was supported by the National Natural Science Foundation of China (Grant numbers 32171208 and 31870750 to H.Z.); by Natural Science Foundation of Tianjin [Grant number 20JCYBJC01320 to J.L.]; by Shenzhen Science and Technology program (Grant number JCYJ20210324122212034 to J.L.); and by Fundamental Research Funds for the Central Universities, Nankai University (Grant number 030/63211052 to J.L.).

## Author contributions

Y.C., H.Z., and J.L. designed the research. Y.C., Z.S., F.C., Y.G., H.X., Q.M., X.C., C.-L.P., T.T.M., and H.Z. performed research. Y.C., H.Z., and J.L. analyzed the data and prepared the figures. Y.C., B.E.P., J.H., H.Z., and J.L. wrote the manuscript. All authors reviewed the manuscript. J.L. coordinated the research.

## Competing interests

The authors declare no competing interests.
