## [Peer Review File · Nature Communications]

REVIEWER COMMENTS

Reviewer #1 (Remarks to the Author):

In this manuscript, Cheng et al. report that SS18's QPGY domain drives liquid-liquid phase separation (LLPS) of the protein and recruits BRG1 to the same phase-separated condensates. These findings provide validation of a recently published Nature Communications article (Kuang et al. PMID: 34215745, listed as Reference #31 in the current manuscript version) that first reported the phase separation properties of SS18.

Cheng et al. further show that the SS18-SSX fusion oncoprotein in synovial sarcoma also retains the ability to undergo LLPS and recruit BRG1 into condensates, and that these properties contribute to the fusion protein's transformation capacity. This data will be of interest to the sarcoma research community as many sarcoma fusion oncoproteins, especially the FET oncoproteins, have increasingly been implicated in LLPS-mediated gene regulation. The relative contributions of SSX vs SS18 to LLPS in SS has been an open question, that the authors have probably settled.

Finally, the authors provide the crystal structure of the human SS18/BRG1 and yeast SNF11/SNF2 heterodimers. While SNF11 is generally regarded as a yeast-specific SWI/SNF subunit, the authors suggested SNF11 as a homolog of SS18 based on similarities in the crystal structures, and if correct will be an important insight to the field. I suspect some of the recently developed AI/computational structure prediction data will soon be able to confirm or refute this postulate.

Major comments:

(1) re Fig 5: SW982 cells are not synovial sarcoma cells - they do not carry SS18-SSX and are derived from a high-grade transformation event in cultured synovial cells, whereas synovial sarcoma as a human disease is not derived from synovium at all. It is well known to the synovial sarcoma research community that this line is mislabelled by ATCC and cannot be used for synovial sarcoma research except as a negative control -- it is a soft tissue sarcoma derived from synovium, but not a synovial sarcoma! The authors will need to repeat these experiments in a bona fide synovial sarcoma cell line such as any of HS-SY-II, SYO-1, Fuji, Aska, Yamato, Mojo, CME-1, or primary explanted human synovial sarcomas.

(2) On the topic of cell line models, even better would be to perform validation in a system that does not require non-physiologic overexpression (endogenous promoters), especially since LLPS is concentration-dependent, although the difficulties of assessing LLPS in native state systems are acknowledged (e.g. McSwiggen et al. 2019 PMID: 31594803). The authors should at least acknowledge these limitations and include these technical caveats when stating their conclusions.

(3) There are errors in multiple figures:

- Fig 3G: The red box in the "pre-bleach" image is mis-placed (should be down and to the right of the current). This error becomes evident when zooming in on the image, reflecting that these images are probably too small to be convincing and need to be shown zoomed in more.
- Fig S5D: Both images are identical. Was the 2nd image supposed to show the superimposed structure from a different angle?
- Figs S6B was not mentioned at all in the text. Figs S5E and S5F (data on SNF11-QPGY

chimera) were only mentioned in the discussion. Similarly, Fig S6C was also mentioned only in the discussion but this relevant data about SSX should be part of the main text in the results section.

- Fig 6 legend to panels F and G are mislabelled.

- The legends for Fig 3, 4 and 5 incorrectly use the unit μM (concentration) for distance, which should be μm .

(4) Introduction: "BAF complex" is a bit too general here; the authors should specify that they are referring to canonical BAF (CBAF) in all places where what they write does not apply to PBAF or GBAF/ncBAF. This distinction has been recognized as quite important in the recent (2019-2021) sarcoma epigenomics and synovial sarcoma literature.

(5) The authors should engage in a more detailed discussion about comparing and relating their manuscript to Reference 31 with regards to the data on SS18 LLPS and BRG1 recruitment

Minor comments:

(1) A thorough edit for written English is needed as there are multiple small errors scattered throughout.

(2) Last sentence of 1st paragraph needs significant rewording (the words "translation" and "generate" are incorrect here).

(3) Fig 1/S1: does the failure to obtain a crystal structure from two separate chains, but success with the single chain, support a LLPS physiology?

(4) Fig S1E is a bit difficult to interpret / understand What do the numbers refer to? What are the different colored dots? Why are Q176 and Q183 seemingly the same dot?

Reviewer #2 (Remarks to the Author):

Cheng et al. applied a series of methods including size exclusion chromatography, analytical ultracentrifugation and X-ray crystallography demonstrating that human SS18 and BRDG1 form a heterodimeric complex. They also show that the yeast homolog of BRG1 forms a highly similar complex with SNF11 through crystal structure determination. Specific hydrogen bonding interactions along with hydrophobic cores were identified as critical elements in complex formation. Further studies revealed that the low-complexity sequence domain of SS18 and the SS18-SSX oncoprotein leads to liquid-liquid phase separation in vitro and in vivo through tyrosine residues mediated self-association, a process important

for the transformation activity of SS18-SSX. The manuscript is well written, and the conclusions drawn are sound.

Minor points:

Fig. 1C: Please indicate the position of the three hydrophobic cores and the hydrogen bonding interactions in the overall structure of the BRG1/SS18 complex (within Fig. 1 or as a supplementary figure).

Supplementary Table S1: The space group for data set "SNF11/SNF2 Native" is listed as C121, whereas the Structure Validation Report states P212121. Please correct.

Please include the PDB entry codes of SS18/BRDG1 and SNF11/SNF2 in Table S1 and in the caption of Figs. 1B and 2B.

Reviewer #3 (Remarks to the Author):

Cheng et al. set out to understand how structural and non-structured elements of the oncogenic fusion protein SS18-SSX fusions could induce cell transformation. A clue was provided by the fact that these fusions form visible puncta in synovial sarcoma cells. Furthermore, SS18 contains a low complexity domain having composition rich in glutamine, proline, glycine, and tyrosine (the QPGY domain) typical of the Prion-Like Domain PLDs found in other oncogenic fusion proteins such as FET (sarcoma, FUS; Ewings sarcoma, EWS; and TAF15). These PLDs are known to undergo liquid-liquid phase separation to form condensates, both in vivo and in vitro, and a growing body of evidence suggests that LLPS is essential to driving oncogenic transformation. The authors solved the structure of key heteromeric complexes of SS18/BRG1 and yeast SNF11/SNF2; homologous complexes of SWI/SNF chromatin remodelling complexes and showed that they are highly similar in structure. This allowed them to design a mutant of SS18 called M3, containing three mutations at the interface of the SS18/BRG1 complex that disrupts their interaction. They further designed mutants of the QPGY domain in which 19 or 21 tyrosines were mutated to serines (Y19 or Y21). The reasoning here was that like PLDs, LLPS could be driven by multivalent aromatic-aromatic interactions with further material properties of resulting condensates determined by the compositions of intervening sequences. The authors observed that SS18-SSX indeed undergoes LLPS in vitro in the presence of 10% Ficoll, that BRG1 partitions into the condensates, but not condensates composed of the SS18 M3 mutant and that the Y19 mutant does not undergo LLPS. Results with these mutations in cells recapitulated the in vitro results. Finally, both M3 and Y19 mutations prevented oncogenic transformation, suggesting that both LLPS and structural interactions of SS18-SSX and BRG1 are essential to oncogenesis.

Overall the studies are well conceived and controlled and the manuscript is written fairly well. The study is a contribution to the growing evidence for the role of LLPS in oncogenesis that has appeared in the literature in the past few years. As such, I don't know whether it is appropriate to a general interest journal like Nature Communications since the principals are already very well documented.

Some improvements could be made in the writing where there are odd object-noun/pronoun relationships. Just for example, in the Abstract, "...SS18-SSX is present in virtually 100% of synovial sarcomas, being the only cytogenetic aberration in most of these tumors". How can a protein be a cytogenetic aberration? Do the authors mean the puncta formed by SS18-SSX?

Another example: "...assemble into heterodimers with a similar model, revealing that SNF11 might be the cognate protein of SS18 in chromatin remodeling complex". Do the authors mean, "similar structure"? I have no idea what a protein being the "cognate protein" of another protein means. Do the authors mean homologue? Structural homologue?

Point-by-point responses to the reviewers' comments:

(Our responses are in blue and all changes in the revised manuscript text file with track changes)

REVIEWER COMMENTS

Reviewer #1 (Remarks to the Author):

In this manuscript, Cheng et al. report that SS18's QPGY domain drives liquid-liquid phase separation (LLPS) of the protein and recruits BRG1 to the same phase-separated condensates. These findings provide validation of a recently published Nature Communications article (Kuang et al. PMID: 34215745, listed as Reference #31 in the current manuscript version) that first reported the phase separation properties of SS18.

Cheng et al. further show that the SS18-SSX fusion oncoprotein in synovial sarcoma also retains the ability to undergo LLPS and recruit BRG1 into condensates, and that these properties contribute to the fusion protein's transformation capacity. This data will be of interest to the sarcoma research community as many sarcoma fusion oncoproteins, especially the FET oncoproteins, have increasingly been implicated in LLPS-mediated gene regulation. The relative contributions of SSX vs SS18 to LLPS in SS has been an open question, that the authors have probably settled.

Finally, the authors provide the crystal structure of the human SS18/BRG1 and yeast SNF11/SNF2 heterodimers. While SNF11 is generally regarded as a yeast-specific SWI/SNF subunit, the authors suggested SNF11 as a homolog of SS18 based on similarities in the crystal structures, and if correct will be an important insight to the field. I suspect some of the recently developed AI/computational structure prediction data will soon be able to confirm or refute this postulate.

We thank the reviewer for nicely summarizing the key findings of our work and for the constructive critiques and suggestions below.

Major comments:

(1) re Fig 5: SW982 cells are not synovial sarcoma cells - they do not carry SS18-SSX and are derived from a high-grade transformation event in cultured synovial cells, whereas synovial sarcoma as a human disease is not derived from synovium at all. It is well known to the synovial sarcoma research community that this line is mislabelled by ATCC and cannot be used for synovial sarcoma research except as a negative control - - it is a soft tissue sarcoma derived from synovium, but not a synovial sarcoma! The authors will need to repeat these experiments in a bona fide synovial sarcoma cell line such as any of HS-SY-II, SYO-1, Fuji, Aska, Yamato, Mojo, CME-1, or primary explanted human synovial sarcomas.

We thank the reviewer to point this error out and for his/her insightful suggestion. It is known that synovial sarcoma cell lines are not commercially available. Accordingly, we collaborated with Prof. Chang-liang Peng from Shandong University, China and Prof. Jian Hu from the University of Texas MD Anderson Cancer Center, and repeated these experiments in synovial sarcoma cell lines HS-SY-II and CME-1. The new acquired data are shown in Fig. 5f, Fig. 5h, Supplementary Fig. 4c, and Supplementary Fig. 4d (below) and the manuscript has been revised accordingly.

Consistently, we found that both SS18-SSX1 and the mutant SS18(3M)-SSX1 show a more dense puncta-like cluster, distributed in both synovial sarcoma cells HS-SY-II (Fig. 5f) and CME-1 (Supplementary Fig. 4c). In contrast, the mutant SS18(Y19S)-SSX1 diffused in both synovial sarcoma cells and a small amount of protein aggregation was observed (Fig. 5f and Supplementary Fig. 4c).

The Mutant SS18(3M)-SSX1 proved its incapability of colocalizing with BRG1, whereas SS18-SSX1-WT was able to recruit BRG1 into the phase-separated condensates in both synovial sarcoma cell-line HS-SY-II (Fig. 5h) and CME-1 (Supplementary Fig. 4d). Additionally, mutant SS18(Y19S)-SSX1, which has the inability of phase separation, was observed to have a diffused distribution of BRG1 in the cell lines as well (Fig. 5h and Supplementary Fig. 4d).

(Fig. 5f) Live-cell imaging for GFP-SS18-SSX1, GFP-SS18(3M)-SSX1, and GFP-SS18(Y19S)-SSX1 in HEK293T and HS-SY-II cells. The scale bar indicates 5 μ m.

Supplementary Fig. 4c

(Supplementary Fig. 4c) Live-cell imaging for GFP-SS18-SSX1, GFP-SS18(3M)-SSX1, and GFP-SS18(Y19S)-SSX1 in CME-1 cells. The scale bar indicates 5 μm.

Fig.5h

(Fig. 5h) Co-expression of GFP-SS18-SSX1 or mutants and Cherry-BRG1 in HEK293T and HS-SY-II cells. The scale bar indicates 5 μm.

Supplementary Fig. 4d

(Supplementary Fig. 4d) Co-expression of GFP-SS18-SSX1 or mutants and Cherry-BRG1 in CME-1 cells. The scale bar indicates 5 μ m.

(2) On the topic of cell line models, even better would be to perform validation in a system that does not require non-physiologic overexpression (endogenous promoters), especially since LLPS is concentration-dependent, although the difficulties of assessing LLPS in native state systems are acknowledged (e.g. McSwiggen et al. 2019 PMID: 31594803). The authors should at least acknowledge these limitations and include these technical caveats when stating their conclusions.

We agree with the reviewer's comments. In current study, we evaluated phase separation of SS18 or SS18-SSX1 in live cells using ectopic overexpression systems (Figs. 3, 4, and 5). In addition, we noted that the endogenous SS18 or SS18-SSX1 shows as a similar puncta-like distribution in mouse embryonic stem or synovial sarcoma cells as that of overexpression in HeLa, HEK293T, or synovial sarcoma cells (Refs. 30 and 32 in the revised manuscript) (Figs. 3, 4, and 5). It is an undeniable that, however, LLPS is a concentration-dependent process. Although it is difficult to assess LLPS in native state systems (Ref. 33 in the revised manuscript), future studies are needed to validate SS18 or SS18-SSX1 LLPS in an appropriate in vivo system. Following reviewer's suggestion, all this information has been included in the second paragraph of the Discussion section from the revised manuscript.

Reference:

30. dos Santos, N. R., *et al.* Nuclear localization of SYT, SSX and the synovial sarcoma-associated SYT-SSX fusion proteins. *Hum. Mol. Genet.* **6**, 1549-1558 (1997).
32. Kuang, J., *et al.* SS18 regulates pluripotent-somatic transition through phase separation. *Nat. Commun.* **12**, 4090 (2021).
33. McSwiggen, D. T., Mir, M., Darzacq, X. & Tjian, R. Evaluating phase separation in live cells: diagnosis, caveats, and functional consequences. *Genes Dev.* **33**, 1619-1634 (2019).

(3) There are errors in multiple figures:

- Fig 3G: The red box in the “pre-bleach” image is mis-placed (should be down and to the right of the current). This error becomes evident when zooming in on the image, reflecting that these images are probably too small to be convincing and need to be shown zoomed in more.

We thank the reviewer to point this error out. Following the reviewer’s suggestion, the zoomed red box area is shown in the lower left corner of each image (revised Fig. 3g, below).

Fig.3g

(Fig. 3g) Representative time-lapse FRAP images showing that GFP-SS18 signal within the puncta recovered within a few seconds in HeLa cells. Red boxes show the zoomed-in regions.

- Fig S5D: Both images are identical. Was the 2nd image supposed to show the superimposed structure from a different angle?

Thanks for pointing this issue out. The original legend for Supplementary Fig. 5d

may cause ambiguity. Actually, Supplementary Fig. 5d are shown in wall-eye stereo mode. This is a standard method for displaying stereo images in publications as it works well when the display (in this case, the piece of paper) is close to the eyes. We have revised the legend as follows: Stereoview by wall-eye mode showing structure comparison of yeast SNF11⁽³⁸⁻¹⁶⁹⁾ (orange)/SNF2⁽²⁴⁸⁻³⁰⁸⁾ (green) complex and human SS18⁽¹⁴⁻¹⁰¹⁾ (cyan)-BRG1⁽¹⁷²⁻²¹³⁾ (magenta) complex.

- Figs S6B was not mentioned at all in the text. Figs S5E and S5F (data on SNF11-QPGY chimera) were only mentioned in the discussion. Similarly, Fig S6C was also mentioned only in the discussion but this relevant data about SSX should be part of the main text in the results section.

Thanks for pointing this issue out. We have added the relevant description of Supplementary Fig. 6b on page 13 in the revised manuscript as follows: “An IUPred analysis indicated that the carboxyl-region, including domains QPGY and RD, of SS18-SSX1 is intrinsically disordered as well (Supplementary Fig. 6b)”.

Following the reviewer's suggestion, the results of Supplementary Figs. 5e, 5f, and 6c were mentioned in the main text of the results sections.

We have added the description of Supplementary Figs. 5e and 5f on pages 11 and 12 in the revised manuscript as follows: “To further examine the effect of tyrosine residues in phase separation, the QPGY domain of SS18 was fused to the carboxyl-terminal of SNF11, and this chimera was named SNF11-QPGY. Supplementary Figs. 5e and 5f provide several lines of evidence demonstrating that SNF11-QPGY chimera might undergo phase separation via QPGY domain-mediated self-association as well”.

We have added the description of Supplementary Fig. 6c on pages 13 and 14 in the revised manuscript as follows: “However, we found that purified SSX1⁽¹¹¹⁻¹⁸⁸⁾ itself cannot undergoes LLPS under the same conditions as SS18-SSX1 protein (Supplementary Fig. 6c)”.

- Fig 6 legend to panels F and G are mislabelled.

Thank you for noticing this mistake. We have corrected this mistake in the revised manuscript.

- The legends for Fig. 3, 4 and 5 incorrectly use the unit μM (concentration) for distance, which should be μm .

Thank you for noticing this mistake. We have changed unit μM into μm in the legends for Fig. 3, 4 and 5 in the revised manuscript.

(4) Introduction: "BAF complex" is a bit too general here; the authors should specify that they are referring to canonical BAF (CBAF) in all places where what they write does not apply to PBAF or GBAF/ncBAF. This distinction has been recognized as quite important in the recent (2019-2021) sarcoma epigenomics and synovial sarcoma literature.

We fully agree with the reviewer's comments. Following the reviewer's suggestion, we have revised manuscript accordingly.

(5) The authors should engage in a more detailed discussion about comparing and relating their manuscript to Reference 31 with regards to the data on SS18 LLPS and BRG1 recruitment.

In the Ref. 32 from the revised manuscript, the authors found that a C-terminal intrinsically-disordered-region of SS18 regulates its forming microscopic condensates under the condition of ectopic overexpression, and that the N-terminal 70 amino acids of SS18 is required for its binding to CBAF. These observations are consistent with our results. Moreover, we used purified recombinant proteins to confirm SS18 LLPS in vitro and revealed the structural basis of the interaction between SS18 and BRG1. Following the reviewer's suggestion, we have added all these descriptions at the end of the first paragraph of the Discussion section from the revised manuscript.

Minor comments:

(1) A thorough edit for written English is needed as there are multiple small errors scattered throughout.

The revised manuscript has been thoroughly edited to eliminate multiple small errors, and we have also done language polishing by native English speaker (e.g. the author Bridgitte E. Palacios). We really hope that the language readability has been substantially improved.

(2) Last sentence of 1st paragraph needs significant rewording (the words "translation" and "generate" are incorrect here).

Thanks so much for pointing these errors out. We have reworded the last sentence of 1st paragraph in the Introduction section as follows: “Contrasting with conventional translocations in other soft tissue sarcomas, the oncofusion protein SS18-SSX lacks any DNA binding domain and is thought to exert its activity by combining with other chromatin modifiers.”

(3) Fig 1/S1: does the failure to obtain a crystal structure from two separate chains, but success with the single chain, support a LLPS physiology?

The protein fragment SS18⁽¹⁴⁻¹⁰¹⁾ resolved in the structure (absent of QPGY domain) does not mediate the phase separation of SS18, and thus we believed that the N-terminal fragment SS18⁽¹⁴⁻¹⁰¹⁾ is not related to LLPS. To avoid confusion, we deleted this statement in the revised manuscript.

(4) Fig S1E is a bit difficult to interpret / understand. What do the numbers refer to? What are the different colored dots? Why are Q176 and Q183 seemingly the same dot?

Thank the reviewer for pointing this issue out. Original Fig. S1E was replaced by Supplementary Fig. 1h (below) in the revised manuscript. The hydrogen-bonding interactions between SS18 and BRG1 are generated by the software Ligplot. We have added the explanation of the symbols in the revised legend of Supplementary Fig. 1h for better understanding.

Supplementary Fig. 1h

(Supplementary Fig.1h) Ligplot diagram in the black frame indicates hydrogen-bonding interactions between SS18 and BRG1. Hydrogen bonds are shown as black dotted lines. The numbers above the lines represent hydrogen bond distance, and the unit is Å. Black solid dots represent carbon atoms, blue solid dots represent nitrogen atoms, and red solid dots represent oxygen atoms.

Reviewer #2 (Remarks to the Author):

Cheng et al. applied a series of methods including size exclusion chromatography, analytical ultracentrifugation and X-ray crystallography demonstrating that human SS18 and BRDG1 form a heterodimeric complex. They also show that the yeast homolog of BRG1 forms a highly similar complex with SNF11 through crystal structure determination. Specific hydrogen bonding interactions along with hydrophobic cores were identified as critical elements in complex formation. Further studies revealed that the low-complexity sequence domain of SS18 and the SS18-SSX oncoprotein leads to

liquid-liquid phase separation in vitro and in vivo through tyrosine residues mediated self-association, a process important for the transformation activity of SS18-SSX. The manuscript is well written, and the conclusions drawn are sound.

We thank the reviewer's appreciation of our work and for the constructive suggestions below.

Minor points:

Fig. 1C: Please indicate the position of the three hydrophobic cores and the hydrogen bonding interactions in the overall structure of the BRG1/SS18 complex (within Fig. 1 or as a supplementary figure).

Original Fig. 1C and Fig. S1E were replaced by Fig. 1c-e and Supplementary Fig. 1h, respectively (below). According to the reviewer's suggestion, the Fig.1c-e (three hydrophobic cores) and Supplementary Fig.1h (hydrogen-bonding interactions) have been revised accordingly. In addition, original Fig. 2C was replaced by the revised Fig. 2c (below).

Fig. 1c-e

(Fig. 1c-e) Ligplot diagrams in the black frame indicate details of hydrophobic interaction between BRG1 and SS18. Three groups of hydrophobic cores are shown as spoked arcs, respectively in c, d, and e.

Supplementary Fig. 1h

(Supplementary Fig.1h) Ligplot diagram in the black frame indicates hydrogen-bonding interactions between SS18 and BRG1. Hydrogen bonds are shown as black dotted lines. The numbers above the lines represent hydrogen bond distance, and the unit is Å. Black solid dots represent carbon atoms, blue solid dots represent nitrogen atoms, and red solid dots represent oxygen atoms.

(Fig. 2c) Ligplot diagram in the black frame indicates details of hydrogen-bonding interactions between SNF11 and SNF2. Hydrogen bonds are shown as black dotted lines. The numbers above the lines represent hydrogen bond distance, and the unit is Å. Black solid dots represent carbon atoms, blue solid dots represent nitrogen atoms, and red solid dots represent oxygen atoms.

Supplementary Table S1: The space group for data set “SNF11/SNF2 Native” is listed as C121, whereas the Structure Validation Report states P212121. Please correct.

Thank you for noticing this mistake. We have corrected this mistake in the revised Supplementary Table 1.

Please include the PDB entry codes of SS18/BRDG1 and SNF11/SNF2 in Table S1 and in the caption of Figs. 1B and 2B.

According to the reviewer's suggestion, we have added the PDB entry codes 7VRB and 7VRC in Supplementary Table 1 or the legends of Fig. 1b and Fig. 2b.

Reviewer #3 (Remarks to the Author):

Cheng et al. set out to understand how structural and non-structured elements of the oncogenic fusion protein SS18-SSX fusions could induce cell transformation. A clue was provided by the fact that these fusions form visible puncta in synovial sarcoma cells. Furthermore, SS18 contains a low complexity domain having composition rich in glutamine, proline, glycine, and tyrosine (the QPGY domain) typical of the Prion-Like Domain PLDs found in other oncogenic fusion proteins such as FET (sarcoma, FUS; Ewings sarcoma, EWS; and TAF15). These PLDs are known to undergo liquid-liquid phase separation to form condensates, both in vivo and in vitro, and a growing body of evidence suggests that LLPS is essential to driving oncogenic transformation. The authors solved the structure of key heteromeric complexes of SS18/BRG1 and yeast SNF11/SNF2; homologous complexes of SWI/SNF chromatin remodelling complexes and showed that they are highly similar in structure. This allowed them to design a mutant of SS18 called M3, containing three mutations at the interface of the SS18/BRG1 complex that disrupts their interaction. They further designed mutants of the QPGY domain in which 19 or 21 tyrosines were mutated to serines (Y19 or Y21). The reasoning here was that like PLDs, LLPS could be driven by multivalent aromatic-aromatic interactions with further material properties of resulting condensates determined by the compositions of intervening sequences. The authors observed that SS18-SSX indeed undergoes LLPS in vitro in the presence of 10% Ficoll, that BRG1 partitions into the condensates, but not condensates composed of the SS18 M3 mutant and that the Y19 mutant does not undergo LLPS. Results with these mutations in cells recapitulated the in vitro results. Finally, both M3 and Y19 mutations prevented oncogenic transformation, suggesting that both LLPS and structural interactions of SS18-SSX and BRG1 are essential to oncogenesis.

Overall the studies are well conceived and controlled and the manuscript is written fairly well. The study is a contribution to the growing evidence for the role of LLPS in oncogenesis that has appeared in the literature in the past few years. As such, I don't know whether it is appropriate to a general interest journal like Nature Communications since the principals are already very well documented.

We sincerely thank the reviewer for nicely summarizing the key findings of our work, and for the suggestions below for improving the language of this manuscript.

Some improvements could be made in the writing where there are odd object-noun/pronoun relationships. Just for example, in the Abstract, "...SS18-SSX is present in virtually 100% of synovial sarcomas, being the only cytogenetic aberration in most

of these tumors”. How can a protein be a cytogenetic aberration? Do the authors mean the puncta formed by SS18-SSX?

Thank the reviewer for pointing these issues out. The revised manuscript has been thoroughly edited to eliminate multiple small errors, and we have also done language polishing by native English speaker (e.g. the author Bridgitte E. Palacios). We really hope that the language readability has been substantially improved.

As to this example, we have changed these descriptions into “Oncoprotein SS18-SSX is a hallmark of synovial sarcomas” and “...resulting in an in-frame fusion gene *SS18-SSX*. This remarkably translocation is present in virtually 100% synovial sarcomas and is often the only cytogenetic aberration” in the revised Abstract and Introduction, respectively.

Another example: “...assemble into heterodimers with a similar model, revealing that SNF11 might be the cognate protein of SS18 in chromatin remodeling complex”. Do the authors mean, “similar structure”? I have no idea what a protein being the “cognate protein” of another protein means. Do the authors mean homologue? Structural homologue?

In this manuscript, we depict the structures of both human SS18/BRG1 and yeast SNF11/SNF2 subcomplexes. Both subcomplexes assemble into heterodimers that share a similar conformation, suggesting that SNF11 might be a homologue of SS18. We have changed the statement “cognate protein” into “homologue” in the revised manuscript.

REVIEWERS' COMMENTS

Reviewer #1 (Remarks to the Author):

All my comments (as Reviewer #1) have been appropriately addressed.

Point-by-point responses to the reviewers' comments:
(Our responses are in blue)

REVIEWERS' COMMENTS

Reviewer #1 (Remarks to the Author):

All my comments (as Reviewer #1) have been appropriately addressed.

Thank you for your support for the publication of our work.